# Layer-specific cortical dynamics during transitions from error monitoring to decision execution in reversal learning

E. Acun[1], M. Zempeltzi[1,4], K. E. Deane[1,5], M. Deliano[1], V. Kannan ®[1,6], F. W. Ohl[1,2] & M. F. K. Happel ®[1,3] ✉

Cognitive flexibility, critical for adaptive behavior, relies on dynamic neural processes across cortical layers. This study investigates the layer-specific temporal and spectral dynamics of the primary auditory cortex (A1) during multiple reversal learning tasks in Mongolian gerbils. Behavioral analysis revealed rapid adaptation across reversal phases, highlighting significant improvements in auditory discrimination and cognitive flexibility over weeks of training. Chronic current source density (CSD) recordings demonstrated distinct synaptic activity patterns, with deep-layer dominance during early learning and increasing superficial layer engagement as performance improved, indicating layer-specific and refined neural signatures linked to behavioral performance. Spectral power analysis demonstrated how error processing and decision-making evolved across different layers and frequency bands. During the reversal phase, prestimulus spectral activity in layer I/II reflected early cortical engagement. In the learning phase, stimulus-locked beta and gamma oscillations in upper and deeper layers were consistent with the integration of sensory input and behavioral output. The retrieval phase was marked by beta and gamma activity before and after stimulus presentation, predominantly in middle and upper layers, supporting refined decision-making. These findings provide insights into how cortical circuit dynamics may support adaptive learning and contribute to the neural basis of cognitive flexibility and layer-specific plasticity in sensory decision-making.

While traditionally viewed as a purely sensory area, the primary auditory cortex (A1) integrates bottom-up acoustic features with top-down task demands. Rodent studies have demonstrated that A1 neurons encode not only sound frequency and amplitude[1], but also abstract task rules and choice-related variables[2]. However, how such cognitive processes are distributed across cortical layers during learning remains a matter of current research[3–6]. Laminar studies suggest that supragranular circuits contribute to attentional gain and connectivity shifts during decision-making[7], underscoring the need to examine how different layers reorganize during adaptive behavior.

Reversal learning requires the suppression of previously learned associations and the establishment of new response contingencies, a process distinct from initial discrimination learning. Thereby, reversal learning can help to shed light on cognitive flexibility and adaptive behavior, engaging multiple brain regions, including the hippocampus, prefrontal cortex, and striatum[8]. Recent studies have challenged the notion that these tasks primarily measure response inhibition, revealing unique features of reversal

learning compared to initial discrimination[9]. Reversal learning is characterized by longer choice latencies after incorrect trials, reduced win-stay/lose-shift strategies, and increased perseveration compared to initial learning[10]. In particular, cortical dopamine circuits in areas such as the prefrontal cortex regulate striatal dopamine dynamics and influence reversal learning processes[11]. Key regions like the dorsolateral prefrontal cortex, dorsomedial prefrontal cortex, and inferior frontal gyrus exhibit overlapping activity during error-feedback processing across different reversal learning conditions[12]. Functional connectivity patterns in the brain change progressively during spatial learning and differ significantly between acquisition and reversal phases[13]. The use of advanced spatial strategies correlates positively with extensive telencephalic connectivity, while non-spatial strategies show negative correlations[13].

Here, we combined multiple-reversal learning with chronic laminar recordings in Mongolian gerbils to investigate how synaptic and spectral activity in A1 changes across phases of adaptive behavior. Specifically, we

[1]Leibniz-Institute for Neurobiology, Magdeburg, Germany. [2]Institute of Biology, Otto-von-Guericke-University, Magdeburg, Germany. [3]MSB Medical School Berlin, Faculty for Medicine, Berlin, Germany. [4]Present address: Fraunhofer Institute for Factory Operation and Automation IFF, Magdeburg, Germany. [5]Present address: University of California, Riverside, Riverside, CA, USA. [6]Present address: Ludwig-Maximilian-University, Faculty of Biology, Munich, Germany. ✉e-mail: max.happel@medicalschool-berlin.de

**Fig. 1 | Surgical procedure for implanting and chronically fixing multichannel electrodes in the primary auditory cortex (A1) of Mongolian gerbils. a** Dorsal view of the gerbil brain (adapted from Luigi Petrucco, SciDraw.io): Animals are secured via a screw nut (gray hexagon) attached to the skull and a metal bar. The blue circle indicates the implantation site to target the right A1 region. The green circle shows the position of the reference electrode on the left side of the skull. Two small screws (yellow circles) are used to increase stability of the implanted device. **b** Lateral view of the brain (adapted from Ann Kennedy, SciDraw.io): A multichannel silicon probe with 32 recording sites was inserted perpendicular to the cortical surface into A1. The electrode, along with the flexible bundle and connector, is chronically fixed onto the skull using UV-curable dental acrylic. Before behavioral training, the dry acrylic is painted with a bright magenta color to enable video tracking of the animal's movements. During experiments, a headstage is connected via a pre-amplifier to acquire local field potential signals from the implanted probe.

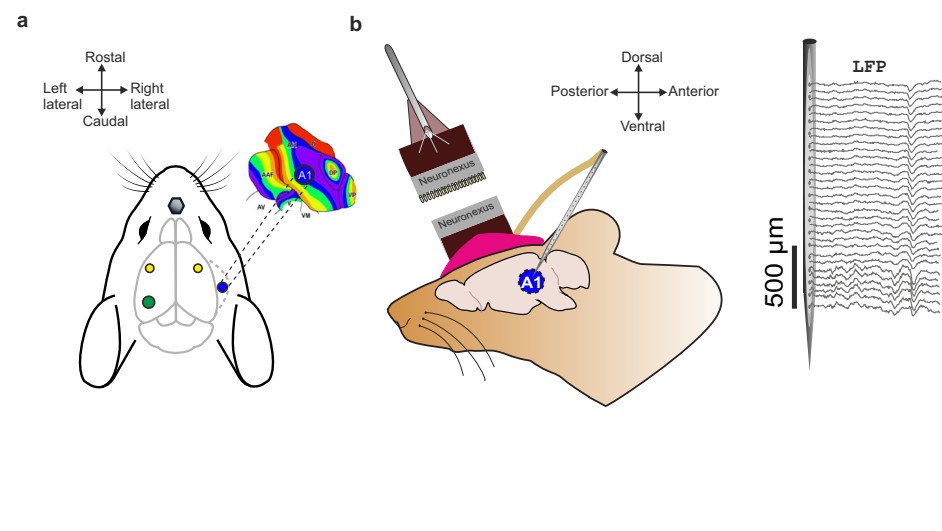

hypothesized that task performance would be accompanied by layer-specific modulations in current source density (CSD), reflecting a redistribution of cortical processing from infragranular to supragranular circuits as learning stabilizes. By linking behavioral adaptation to laminar reorganization, our findings provide new insight into the role of auditory cortex in cognitive flexibility and sensory decision-making.

## Methods

### Statement of compliance with ethical regulations

Experiments were carried out with adult male Mongolian gerbils (*Meriones unguiculatus*, 4–8 months of age, 70–90 g body weight, total *n* = 9). All experiments presented in this study were conducted in accordance with ethical animal research standards defined by the German Law and approved by an ethics committee of the State of Saxony-Anhalt, and we complied with all relevant ethical regulations for animal use.

### The animal model: mongolian gerbil

In this study, we used adult male Mongolian gerbils aged 4–8 months and weighing between 70 and 90 g (total *n* = 8). The gerbils were sourced from in-house breeding colonies and were housed under controlled laboratory conditions with a temperature of 25 °C, humidity of 30–50%, and a 12-h light/dark cycle. The cages (Type IV) were enriched with tunnels, wheels, hay, woodchips, and paper bedding to ensure animal well-being. Before the start of the experiments, all animals underwent a behavioral screening protocol to exclude those with epileptic tendencies, as gerbils have a genetic predisposition to seizures[14]. Only animals that passed this screening were used for the subsequent surgical and training procedures. The gerbils were trained in a two-way active avoidance shuttle-box task, which is a well-established behavioral paradigm for studying operant conditioning and learning. The shuttle-box consists of two compartments separated by a hurdle, allowing the animals to move between compartments in response to auditory stimuli[15,16].

### Surgical preparation and electrode chronic implantation

For chronic in vivo electrophysiological recordings, a multichannel electrode (NeuroNexus, A1x32-6 mm-50-177_H32_21mm) was surgically implanted into the primary auditory cortex (A1) of the right hemisphere. The gerbils were initially anesthetized with an intraperitoneal (i.p.) injection (0.004 ml/g) consisting of 45% ketamine (50 mg/ml, Ratiopharm GmbH), 5% xylazine (Rompun 2%, Bayer Vital GmbH), and 50% isotonic sodium-chloride solution (154 mmol/1, B. Braun AG). Anesthesia was maintained during the surgery with a continuous i.p. infusion of around 0.15 ml/g*h.

The primary auditory cortex (A1) of the right hemisphere was exposed through a small trepanation (Ø 1 mm) in the temporal bone, which avoids tissue damage and ensures a stable fixation of the implanted electrode[17,18]. Another small hole was drilled in the parietal bone on the contralateral side for the reference wire (stainless steel, Ø 200–230 µm). The animals were head-fixed using a screw-nut glued to the rostral part of the exposed nasal bone plate with UV-curing glue (Plurabond ONE-SE and Plurafill flow, Pluradent). The recording electrode with a flexible bundle between the shaft and connector was inserted perpendicular to the cortical surface into A1 via the small hole. During the implantation, the animals were placed in a Faraday-shielded and acoustically soundproofed chamber. To ensure accurate targeting of A1 and functional coverage of the trained frequency range (1–4 kHz), recordings were verified using tone-evoked CSD responses during and after implantation. Functional confirmation of A1 was based on the presence of short-latency current sinks (<20 ms) in granular (layer III/IV) or infragranular layers (layer Vb), consistent with lemniscal thalamo-cortical input[5,16]. While frequency tuning generally appears broader under awake conditions[19,20], all sites showed best frequencies within or adjacent to the behavioral training range and consistent with primary auditory field identity. This procedure has been previously established by Zempeltzi et al.[5] with detailed frequency and onset latency tunings. Before enclosing the exposed A1 with UV-glue, an antiseptic lubricant (KY-Jelly, Reckitt Benckiser-UK) was applied to the cortical surface. At last, the multichannel probe was permanently fixed in place.

After the surgery, the wounds were treated with a local antiseptic tyrothricin powder (Tyrosur, Engelhard Arzneimittel GmbH & Co.KG), and the animals received analgesic treatment with Metacam (Boehringer Ingelheim GmbH) for the next 2 days. After the surgery, the gerbils underwent a recovery period of at least 3 days before the first awake electrophysiological recording session commenced (see Fig. 1).

After the recovery period, animals were placed in a 1-compartment box in an electrically shielded and soundproof chamber in order to re-characterize the tuning properties of the chronically implanted electrode. Acoustic stimuli were presented in a pseudo-randomized order of pure-tone frequencies covering a range of 7 octaves (0.25–16 kHz; tone duration: 200 ms, ISI 800 ms, 50 pseudo-randomized repetitions, sound level 70 dB SPL), while laminar LFP signals were recorded. Sounds were presented from a loudspeaker (Tannoy arena satellite KI-8710-32) 1 m distance to the animal. A measurement microphone and conditioning amplifier were used to calibrate acoustic stimuli (G.R.A.S. 26AM and B&K Nexus 2690-A, Bruel & Kjaer, Germany).

**Fig. 2 | Experimental design of auditory Go/NoGo tasks. a** The two-way avoidance shuttle-box training setup with chronic recordings in behaving Mongolian gerbils. The subjects were trained to respond to two different pure tone frequencies (1 kHz and 4 kHz), serving as the conditioned stimuli (CS) in a Go/NoGo task designed to avoid an unconditioned stimulus (US - mild foot shock). **b** The sequence of consecutive CS within a trial is shown, along with the length of the observation window (6 s), inter-stimulus interval (1.5 s), and potential behavioral choices. The lightning icon indicates the onset of the unconditioned stimulus (US), which is terminated by a compartment change (red arrows). The time point of the compartment change is recorded as reaction time (RT). Time intervals reflect schematic binning rather than exact temporal scaling. **c** Theoretical framework illustrating the signal detection theory used in this study. The d-prime (d') value represents the sensitivity index used to quantify the animals' discrimination performance. Correct Rejections (Corr. Rej.), Hits, False Alarms (FA), and Misses are indicated in the signal strength probability distributions. **d** The training protocol, detailing the phases of initial discrimination and multiple reversals with corresponding performance metrics and showing the progression of learning and adaptability in auditory-guided decision-making tasks. Figure adapted by Zempeltzi et al.[5].

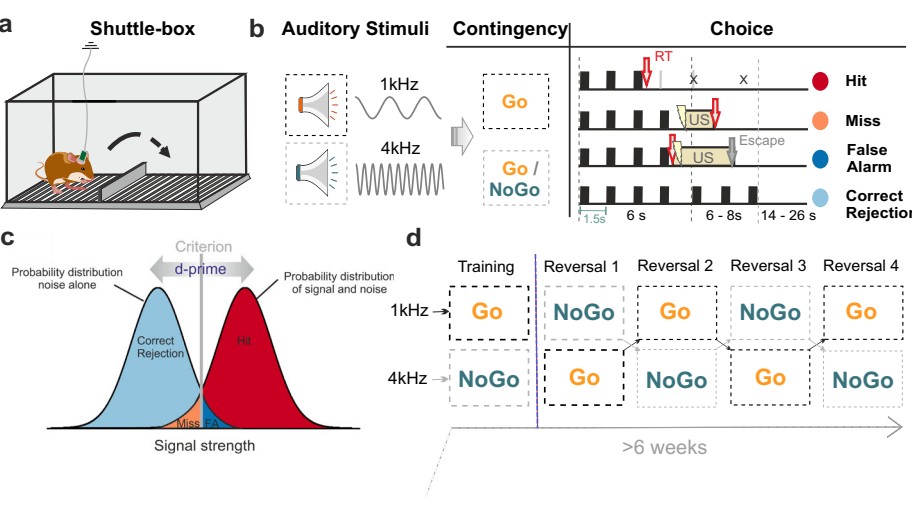

## Chronic laminar recording during a freely-moving shuttle-box task

Operant conditioning was trained using a two-way active avoidance shuttle-box task. The shuttle-box (E15, Coulbourn Instruments) contained two compartments separated by a hurdle (3 cm height), allowing the animals to move between compartments.

Conditioned stimuli (CS) were auditory stimuli generated in Matlab (MathWorks, R2012b), converted to analog by a data acquisition card (NI PCI-6733, National Instruments), routed through an attenuator (gPAH Guger, Technologies), and amplified (Black Cube Linear, Lehman). Two electrostatic loudspeakers (positioned 5 cm at both sides of the shuttle-box) delivered the sounds. A measurement microphone and conditioning amplifier (G.R.A.S. 26AM and B&K Nexus 2690-A, Bruel&Kjaer, Germany) were used to calibrate acoustic stimuli.

Unconditioned stimuli (US) in the form of a mild foot-shock were conditionally delivered through the grid floor and generated by a stimulus generator (STG-2008, Multi-Channel Systems MCS GmbH). The starting intensity of the foot-shock was set to 200 μA. Depending on the individual animal's sensitivity and performance, the shock intensity was adjusted in steps of 50 μA until the escape latencies were below 2 s, in order to achieve a successful association of conditioned stimuli (CS) and US.

Chronic LFP recordings during training were performed by connecting the head-connector of the animal to a preamplifier (20-fold gain, band-pass filtered, HST/32V-G20; Plexon Inc.) and a data acquisition system (Neural Data Acquisition System Recorder/64; Plexon Inc.). The cable harness was wrapped by a metal mesh for bite protection. Tension of the cable was relieved by a spring and a turnable, motorized commutator (Plexon Inc.) that permits free movement and rotation of the animal in the box. To avoid ground loops between the recording system, shuttle-box, and the animal, we ensure proper grounding of the animal via its common ground and leave the grid floor on floating voltage. Broadband signals were recorded continuously using the preamplifier (Plexon REC/64 Amplifier; 1 Hz–6 kHz) during the training with a sampling frequency of 12 kHz. Local field potentials were sampled at 2 kHz, visualized online (NeuroExplorer, Plexon Inc. Recording Controller), and stored offline for further analysis.

## Auditory Go/NoGo shuttle-box task with multiple reversals and data analysis

**Behavioral training protocol.** The gerbils ($n = 8$) were trained using a two-way active avoidance shuttle-box task (E15, Coulbourn Instruments), which contained two compartments separated by a hurdle (3 cm height) allowing the animals to move between compartments (Fig. 2 provides details of the behavioral training and analysis). The conditioned stimuli (CS) were auditory tones generated in Matlab (MathWorks, R2012b), converted to analog by a data acquisition card (NI PCI-6733, National Instruments), and presented via electrostatic loudspeakers positioned 5 cm from both sides of the shuttle-box. The unconditioned stimuli (US) were mild foot-shocks delivered through the grid floor, with the starting intensity set at 200 μA and adjusted in steps of 50 μA to achieve optimal performance (max. 500 μA).

Behavioral training was conducted twice daily with a minimum 5-h interval between sessions. Each session began with a 3-min habituation period. The initial training phase utilized two pure tones (1 kHz and 4 kHz) as CS+, both indicating a 'Go' response, requiring the animals to move between compartments to avoid the US. Upon achieving stable detection performance (d' > 1 for 3 consecutive sessions), the paradigm transitioned to a discrimination task where the 4 kHz tone was reassigned as a 'NoGo' stimulus (CS-). The discrimination phase required the animals to remain in the current compartment to avoid the US when the CS− was presented. Each session contained 60 trials (30 CS+, 30 CS−). Each trial lasted 12-15 s, in which the CS+ tones were repeatedly presented (tone duration 200 ms, ISI of 1.5 s, 70 dB SPL) in a 6 s observation window. A correct response (hit) was defined as a compartment change during this window. If no response occurred, the trial was classified as a miss, and the overlapping CS/US continued until escape behavior was initiated[5].

Successful acquisition of the discrimination task was defined as achieving a sensitivity index (d') above 1.0, reflecting reliable discrimination performance. This criterion was applied prior to initiating reversal phases and also served as the benchmark for the "retrieval" phase in subsequent analyses. Following successful acquisition of the discrimination task, the experiment incorporated multiple reversal phases, inverting the

contingencies of the CS+ and CS−. Each reversal phase lasted one week, challenging the animals' cognitive flexibility and adaptive learning capabilities. Performance metrics included hit rate (correct responses to CS+), false alarm rate (incorrect responses to CS−), and the sensitivity index (d'), calculated as the difference between the z-transforms of hit rate and false alarm rate.

This task requires the suppression of a previously learned response and the acquisition of a new association, making it a robust model for cognitive flexibility. To analyze corresponding learning-related neural dynamics, we divided training data into three distinct performance-based phases. The early reversal phase (d' < 0) was characterized by high levels of perseverative errors, indicating difficulty in adapting to new contingencies. The intermediate learning phase (0 < d' < 1) corresponded to the steepest increase in performance, reflecting active acquisition of the new rule set. The retrieval phase (d' > 1) represented stabilized performance with above-threshold discrimination ability, demonstrating successful learning and retrieval of the newly acquired contingencies. This classification allowed us to distinguish neural correlates of error processing, active learning, and stable retrieval. Each performance bin was supported by substantial trial counts (low: 4244 trials, medium: 4003 trials, high: 5577 trials, distributed across all choice categories), ensuring that the observed differences in CSD activity cannot be attributed to unequal variance or insufficient sampling. Importantly, all animals contributed trials to each performance bin (grand total of 13,824 trials across all bins).

**Electrophysiological recordings**. Chronic laminar local field potential (LFP) recordings were performed using a 32-channel Neuronexus silicon probe implanted in the primary auditory cortex (A1). During training, the head-connector was linked to a preamplifier and a data acquisition system, allowing for continuous broadband signal recording. Signals were sampled at 12 kHz and visualized online, with LFP signals downsampled to 2 kHz for offline analysis.

Raw behavioral data were stored as text files, while electrophysiological data were stored in Plexon format, resulting in a total data volume of 178 GB. Data preprocessing involved converting these files into MATLAB format, resulting in a structured dataset for analysis. A custom graphical user interface (GUI) facilitated the inspection and marking of artifacts, with linear interpolation applied to affected channels and exclusion of trials with significant artifacts.

For behavioral analysis, reaction times, conditioned response rates, and the sensitivity index (d') were calculated. The data were z-normalized for statistical comparisons across sessions and subjects. Current source density (CSD) analysis was performed on the LFP recordings to estimate the spatiotemporal distribution of current sources and sinks across cortical layers. This analysis provided insights into synaptic activity patterns within the cortical microcircuitry. The averaged rectified current source density (AVREC) was calculated to represent the total synaptic activity across all cortical layers. Statistical analyses included one-way repeated measures analysis of variance (rmANOVA) and generalized linear mixed models (GLMM) with logistic regression to predict binary outcomes. Spectral power analysis was conducted using continuous wavelet transform (CWT) to decompose LFP signals into time-frequency representations, focusing on frequency bands from theta to high gamma.

**Current source density (CSD) analysis**

To analyze the laminar local field potentials (LFPs) recorded from the primary auditory cortex, we employed current source density (CSD) analysis. This technique allows for the estimation of the spatiotemporal distribution of current sources and sinks across cortical layers, providing insights into the synaptic activity patterns within the cortical microcircuitry. The CSD was calculated as the second spatial derivative of the recorded field potentials using the following formula:

$$-CSD \approx \delta^2\Phi(z)/\delta z^2 = [\Phi(z + n\Delta z) - 2\Phi(z) + \Phi(z - n\Delta z)]/(n\Delta z)^2$$

whereas Φ represents the field potential; z is the spatial coordinate perpendicular to the cortical laminae; Δz denotes the spatial sampling interval between recording sites; n is the differentiation grid used for the CSD calculation

To enhance the spatial resolution and reduce noise, we applied a spatial smoothing procedure to the LFP profiles. This involved using a weighted average (Hamming window) across 9 channels, corresponding to a spatial kernel filter of approximately 400 µm[16,18]. The resulting CSD distributions reflect the local spatiotemporal flow of ionic currents from extracellular to intracellular space, which are evoked by coordinated synaptic activity within laminar neuronal structures. Specifically, current sinks (negative CSD values) represent the influx of positive ions into neurons, typically associated with excitatory postsynaptic potentials[21,22]. Conversely, current sources (positive CSD values) indicate the efflux of positive ions, often reflecting passive return currents or inhibitory postsynaptic potentials. To describe the overall columnar processing, we calculated the averaged rectified current source density (AVREC) using the following formula:

$$AVREC(t) = \Sigma\left|CSDi(t)\right|/n$$

whereas, n is the number of recording channels; t is time; CSDi represents the CSD value at each channel i

The AVREC reflects the temporal overall local current flow of the columnar activity[23,24] and provides a measure of the total synaptic activity across all cortical layers at each time point. Although initially developed in anesthetized preparations[23], this approach has since been validated in awake animal studies, including in laminar recordings from auditory cortex[5,25]. By analyzing these CSD and AVREC patterns, we can infer the sequence of neural activation across cortical layers, providing a mesoscopic view of the synaptic population activity[16,21]. This approach allows us to investigate how different cortical layers contribute to sensory processing, task-related information encoding, and decision-making processes in the auditory cortex during our behavioral paradigm.

Single-trial data underwent preprocessing using a custom-developed graphical user interface (GUI) implemented in MATLAB (MathWorks, R2016a & R2017b). This GUI allowed for visualization of the local field potential (LFP), current source density (CSD), and behavioral parameters. To that end, channels showing excessive noise (defined as peak-to-peak amplitudes exceeding ±3 standard deviations across trials), persistent line noise, or visible artifacts were first evaluated visually and then excluded from further analysis. This procedure follows established criteria from our prior work using chronic CSD recordings in behaving animals[5].

For affected channels, a linear interpolation method was applied across neighboring, unaffected channels at the LFP level to substitute the compromised data[16]. Shock-induced signal clipping was removed from the overall signals. Trials containing artifacts were excluded from further analysis. This preprocessing step was crucial to ensure data quality and reliability for subsequent analyses, particularly given the long-term nature of the recordings and the potential for various sources of interference in freely behaving animals.

**CSD parameter analysis**. The identification and delineation of cortical layers were based on established methodologies from previous studies[16,19]. In the auditory cortex, early dominant current sinks are indicative of thalamocortical input in granular layers III/IV and infragranular layers Vb/VI. This characteristic pattern allows for the identification of supragranular layers I/II and infragranular layers Va and VI in the CSD recordings (Fig. 3).

For quantitative analysis, we calculated trial-by-trial root-mean-square (RMS) values of averaged CSD traces within each of the five cortical depths. This computation was performed from the tone onset of each CS presentation over a time window of 500 ms. Additionally, the RMS value of the AVREC was calculated within the same time windows to represent the

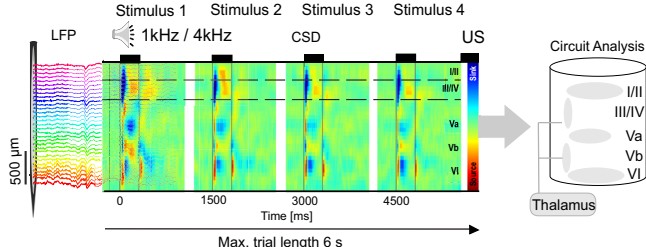

**Fig. 3 | Chronic CSD recordings from the A1 in awake behaving gerbils.** In vivo multichannel LFP recordings were obtained by single-shank (32 channels) silicon probes chronically implanted perpendicular to the surface of the auditory cortex targeting all cortical layers (I–VI). From laminar LFP signals single-trial current source density (CSD) distributions were calculated offline (here shown is a CSD averaged over 30 repetitions). During CS-presentation (200 ms, black frame) tone-evoked CSD components appeared as current sink (in blue) and source (in red) activity reflecting the well-known feedforward information flow of sensory information in the A1 (Happel et al. 2010a). Post-stimulus analysis was performed within a 0–500 ms window following each CS onset, as indicated by the bounding box around each stimulus segment. A simplified schematic illustration of the cortical column and its layers is shown at the right part of the figure. Figure adapted from Zempeltzi et al.[5].

corresponding overall columnar response. It is important to note that time-points after a conditioned response (CR) were not inspected, as the CS presentation was terminated upon CR. To facilitate statistical comparisons, single-trial values were z-normalized across trials. This normalization procedure allows for standardized comparisons of signal parameters across different recording sessions and subjects.

**Statistical analysis of variance**. To assess statistical differences between groups, we employed one-way repeated measures analysis of variance (rmANOVA) using R Studio (R 3.5.1). This approach was chosen to account for the hierarchical structure of the data. We adopted an overall significance level of $\alpha = 0.05$.

For post-hoc comparisons following significant main effects or interactions, we conducted paired-sample $t$-tests with Holm-adjusted significance levels to control for multiple comparisons. Prior to analysis, data were generally z-normalized within each animal and session to facilitate comparisons across subjects and time points.

To quantify effect sizes, we calculated the generalized eta squared ($\eta2gen$) using the R package DescTools. This measure provides an estimate of the proportion of variance explained by each factor or interaction. We interpreted effect sizes according to the following guidelines: $\eta2gen \leq 0.1$ as small, $0.1 < \eta2gen < 0.25$ as medium, and $\eta2gen \geq 0.25$ as large effects.

This statistical approach allowed us to rigorously assess the significance and magnitude of observed differences in cortical activity patterns across experimental conditions while accounting for the repeated measures design and controlling for multiple comparisons.

**Generalized Linear Mixed Model Analysis**. To compare binary outcome classes on a single-trial level, we employed generalized linear mixed-effect models (GLMM) with a logistic link function[26]. GLMM calculations were performed in R Studio (R 3.5.1) using the lme4 package for model estimation, with ggplot2 and sjplot for visualization.

The logistic regression was utilized to predict the probability of the binary (0/1) dependent variables $\pi_i = E(y_i)$. Predictions were transformed by the logistic link function: $g(x) = 1 / (1 + \exp(-x))$ to map model predictions to the interval [0,1].

In the mixed-effects logistic regression, random effects were introduced to account for subject-specific variance:

$$g(E(y_i)) = X_i\beta + Z_iv_i$$

whereas $yi$ is the vector of all responses for the ith animal, $X_i$ and $Z_i$ are design matrices, $\beta$ represents fixed effects, and $v_i$ denotes animal-specific random effects. In our case, $y_i$ represented the z-scored RMS value derived from the CSD traces, modeled as a function of behavioral choice (hit, miss, false alarm, correct rejection), performance level (low, medium, high d′), and cortical layer (I/II, III/IV, Vb, VI), including their interactions. Random intercepts and slopes were included for subject identity to account for inter-individual variability. This structure allowed us to assess how behavioral state and laminar dynamics jointly influenced trial-level neural responses.

The model parameters can be interpreted as logarithmic odds ratios $\log(\pi ij / (1-\pi ij))$, where $\pi ij$ corresponds to the probability of outcome 1 for animal i in trial j. This allows for intuitive interpretation of predicted values (choice probabilities) and estimated coefficients (logarithmic odds ratios).

Random intercepts were incorporated to account for general variability in overall activity across subjects. Random slopes were included to allow fixed effects to vary between animals. AVREC RMS values were z-normalized for the GLMM to facilitate estimation.

Model evaluation utilized the marginal (R2m) and conditional (R2c) coefficients of determination, calculated using the MuMIn package[27]. R2m represents variance in the dependent variable explained by fixed effects, while R2c reflects total variance explained by fixed and random effects. In binary GLMMs, R2m is sample size-independent and dimensionless, enabling comparison across datasets. For each GLMM, trials were pooled across multiple recording sessions, with each animal contributing several hundred valid trials per comparison. Specifically, we conducted GLMMs on the following trial numbers: _Low performance (d′ < 0):_ Hits = 380, FA = 753, CR = 1366, Miss = 1745; _Medium performance (0 ≤ d′ < 1):_ Hits = 769, FA = 418, CR = 1589, Miss = 1237

_High performance (d′ ≥ 1):_ Hits = 1774, FA = 341, CR = 2447, Miss = 1015. Depending on the specific contrast (e.g., behavioral choice, performance level, cortical layer), individual models were generally based on more than 100 trials per condition. This ensured stable estimation of fixed effects while accounting for within-subject variability. Effect sizes were interpreted as small for R2m ≤ 0.1, medium for 0.1 < R2m < 0.25, and large for R2m ≥ 0.25[28].

**Spectral power analysis**. Spectral power analysis was conducted on the CSD signal recorded from the primary auditory cortex (A1) of Mongolian gerbils during the discrimination and reversal phases of the auditory Go/NoGo task. The analysis aimed to compare neural responses associated with different behavioral outcomes: hits, misses, false alarms, and correct rejections.

We applied the continuous wavelet transform (CWT) to CSD signals rather than raw LFPs in order to enhance laminar specificity by isolating transmembrane current sinks and sources. This approach minimizes volume conduction effects and common-mode signals, thereby improving spatial interpretability without compromising the physiological validity of oscillatory power estimates. It is consistent with previous work applying spectral analyses to CSD for laminar oscillations[25]. Unlike traditional time-domain analyses, which focus on the temporal features of neural signals, wavelet-based analysis offers a combined time-frequency approach. This is especially advantageous for non-stationary signals such as the CSD, where neural oscillations vary dynamically over time. By applying wavelet transforms, we can track the evolution of specific frequency bands (e.g., beta and gamma) across time, enabling a more detailed investigation of how different cortical layers contribute to decision-making processes.

CWT was employed using MATLAB (MathWorks, R2016a-R2022a) to decompose LFP signals into time-frequency representations. CWT parameters included frequency limits of 5–100 Hz and utilization of the analytic Morse wavelet, selected for its adjustable time and frequency spread parameters[29–31]. This function uses L1 normalization, where equal amplitude oscillatory components at different scales will have equal magnitude in the CWT. The CWT was applied to CSD signals from recording channels corresponding to different cortical layers. Trial-averaged scalograms were

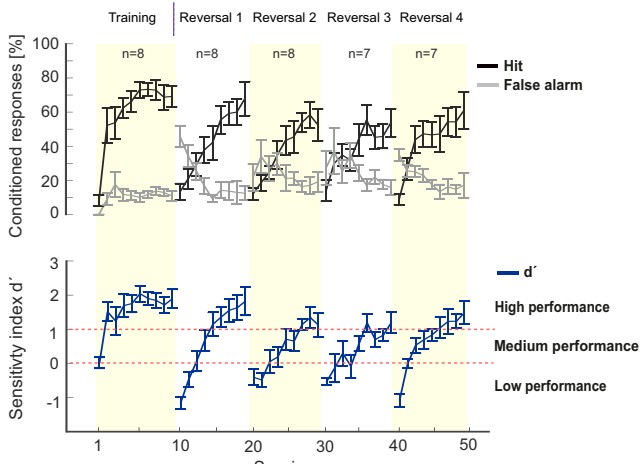

**Fig. 4 | Learning curves during multiple reversal Go/NoGo auditory tasks.** *Top*, Averaged conditioned response rates (mean ± SEM) across initial discrimination and four reversal phases (R1-R4) are shown for hit rates to CS+ (black) and for false alarm rates to CS- (grey). Alternating white and yellow backgrounds indicates the start of each reversal phase where CS contingencies were switched. *Bottom*, Sensitivity index d' calculated from hit and false alarm rates. Horizontal dotted line indicates performance criteria differentiating the early reversal (d' < 0), the intermediate learning phase (0 < d' < 1), and the retrieval phase (d' > 1). The chosen d' boundaries reflect conceptually distinct performance levels (reversal phase, learning phase and robust discrimination), in line with established signal detection theory (see also Methods section). Each session contained 60 trials (30 CS+, 30 CS−).

calculated for each layer. Magnitude of the wavelet transform was computed as:

$$\text{Magnitude} = |a + bi|$$

where a+bi represents the complex output of the trial-averaged CWT analysis[32]. Power was calculated as the square of the magnitude:

$$\text{Power} = |a + bi|^2$$

To assess differences between groups, we employed a non-parametric permutation cluster-mass analysis[33,34]. Student's *t*-test for each time-frequency point across the groups resulted in point-wise *t*-values that were considered significant if they corresponded to a two-tailed *p*-value of less than 0.05. Such significant clusters were identified using the Matlab function *bwboundaries*. Subsequently, we randomly shuffled the group assignments and repeated the *t*-tests and cluster measurements on these permuted groups. This process was executed 1000 times to create a distribution of cluster sizes expected under the null hypothesis. Clusters observed in the actual group comparison that exceeded 95% of the sizes found in the permutation distribution were deemed significant. Clusters were considered significant if their size exceeded the 95th percentile of the null distribution (based on 1000 permutations), using a one-tailed threshold appropriate for positive-valued cluster size statistics. To reduce the likelihood of spurious focal effects, we applied two additional constraints on reported clusters. First, we excluded any clusters smaller than 5 × 5 spatially connected bins in the time–frequency matrix. Second, we computed Cohen's *d* at each bin and retained only those clusters with a mean effect size greater than 0.4. These criteria ensured that all reported clusters reflect both spatially consistent and statistically robust differences between conditions. We now report whether the identified clusters fell within specific frequency bands, which we defined as follows: theta (4–7 Hz), alpha (8–12 Hz), low beta (13–18 Hz), high beta (19–30 Hz), low gamma (31–60 Hz), and high gamma (61–100 Hz).

This approach facilitates the identification of significant group differences while effectively managing multiple comparisons without relying on

assumptions regarding normality or homogeneity of variance. This comprehensive spectral analysis approach provided insights into the neural dynamics underlying cognitive flexibility and decision-making processes in the primary auditory cortex during the auditory Go/NoGo task.

### Statistics and reproducibility

Experiments were conducted on adult male Mongolian gerbils (*n* = 8). Each animal represents one biological replicate. Repeated-measures ANOVAs and generalized linear mixed-effects models (GLMMs) accounted for within-subject structure and inter-individual variability. Holm-adjusted post-hoc tests were applied where appropriate. Effect sizes are reported where relevant. For time–frequency analyses, significance was assessed using cluster-based permutation testing (1000 permutations) with spatial-extent and effect-size criteria to control for multiple comparisons. Each behavioral or neural parameter was derived from multiple sessions and several hundred valid trials per animal. Technical replicates correspond to repeated trials from the same cortical column and were modeled as repeated measures rather than independent observations. All key effects were reproducible across animals.

### Reporting summary

Further information on research design is available in the Nature Portfolio Reporting Summary linked to this article.

## Results

### Multiple Reversals of the choice outcome contingencies

In the initial training phase, subjects were trained to discriminate between two pure tone frequencies—1 kHz and 4 kHz—serving as conditioned stimuli (CS) in a Go/NoGo task (Figs. 2 and 4). The 1 kHz tone was designated as the CS+ (Go stimulus), requiring the animals to move between compartments to avoid the unconditioned stimulus (US), a mild foot shock. Conversely, the 4 kHz tone served as the CS- (NoGo stimulus), where the correct response was to remain in the current compartment to avoid the US. Over the course of the initial discrimination training, gerbils learned to distinguish between the Go and NoGo stimuli, demonstrating significant improvements in performance metrics such as hit rate (correct responses to CS+), false alarm rate (incorrect responses to CS−), and the sensitivity index (d'). Methodologically, this phase was critical for establishing the baseline discrimination performance before introducing the more complex reversal learning tasks. The use of the two-way active avoidance shuttle-box task allowed for precise measurement of the gerbils' ability to learn and adapt to the auditory discrimination task, setting the stage for subsequent analyses of cognitive flexibility during multiple reversal phases. It is important to note that the discrimination phase served as a training phase to ensure that animals could reliably discriminate between the conditioned stimuli (CS+ and CS−). The detailed analysis of this phase has been extensively covered in our previous study[5], which focused on the neural mechanisms underlying auditory discrimination learning. In the current study, we build upon these findings by examining the neural dynamics during multiple reversal learning phases to understand cognitive flexibility.

The overall increase in the sensitivity index (d') across training sessions, with performance stabilizing over time, is reflecting the enhanced discrimination ability between the two tones (Fig. 4). Following successful acquisition of the initial discrimination, the experimental paradigm incorporated multiple reversals of the choice-outcome contingencies over several weeks. In these reversal phases, the stimulus-response associations were switched, such that the previously rewarded stimulus (CS+) became the non-rewarded stimulus (CS−) and vice versa (see Fig. 4). This manipulation challenged the animals' cognitive flexibility, requiring them to adapt their learned responses to the new contingencies. The incorporation of multiple reversal phases allowed for the assessment of the animals' ability to repeatedly update their stimulus-response associations, providing insight into the neural mechanisms

**Fig. 5 | Cortical CSD activity during multiple reversal tasks at different performance levels.**
**a** Representative CSD profile illustrating tone-evoked responses (1st CS presentation, 200 ms duration indicated by dashed vertical lines) for the four behavioral outcomes: hit, miss, false alarm, and correct rejection. Profiles are averaged across reversal sessions to highlight laminar activity patterns associated with each behavioral choice.
**b** Averaged AVREC RMS values (500 ms window stimulus-locked to the onset of the 4th CS cf. Zempeltzi et al.[5]; their Fig. 3a) plotted by behavioral choices (*n* = 8) across all reversal blocks, split by performance level. This window was chosen to capture not only stimulus-evoked activity but also its modulation by behavioral choice. Performance bins were defined according to established thresholds in signal detection theory (d′ = 0 for chance discrimination; d′ = 1 for robust discrimination) and were supported by substantial trial counts (low: 4244 trials, medium: 4003 trials, high: 5577 trials). In low performance (d′ < 0), miss and false alarm trials differ significantly from correct rejection trials. In medium performance (0 < d′ < 1), cortical activity differs significantly across all choices except hit versus false alarm. In high performance (d′ > 1), cortical activity is strongest during correct hits and significantly different from all other choices. Box-plots show median, interquartile range, full data range, and outliers. Significant differences are indicated by a 2-way rmANOVA with Holm-corrected post-hoc tests (*$p \le 0.05$, **$p \le 0.01$, ***$p \le 0.001$).

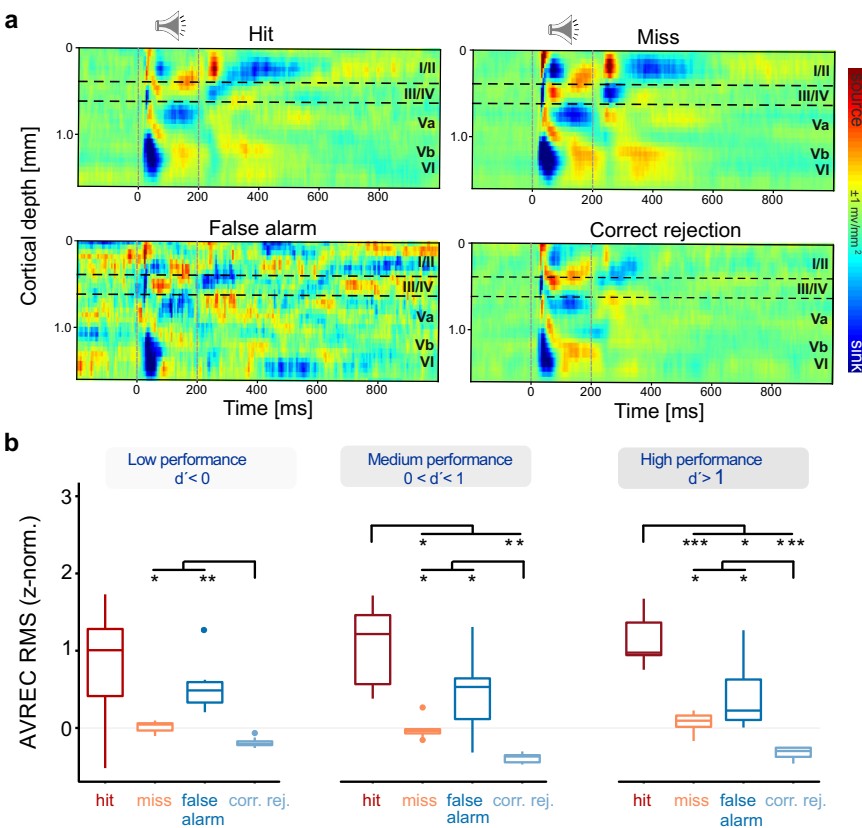

underlying behavioral flexibility in auditory-guided decision making. This experimental design, combining chronic electrophysiological recordings with a complex behavioral paradigm, enables the investigation of neural correlates of auditory discrimination learning and cognitive flexibility in the Mongolian gerbil model.

During the initial discrimination phase, animals demonstrated rapid improvement in their ability to differentiate between the CS+ (Go stimulus) and CS− (NoGo stimulus). The conditioned response rates for hits (correct responses to CS+) increased steadily, reaching approximately 80% by the end of this phase. Concurrently, false alarm rates (incorrect responses to CS−) decreased and stabilized around 20%. This divergence in response rates indicates successful acquisition of the discrimination task. The sensitivity index (d′) also showed marked improvement during this phase. Starting from near-chance levels (d′ ≈ 0), it rapidly increased to values exceeding 1, typically within the first few sessions. This improvement in d′ values reflects enhanced discriminability between the two tones and suggests effective learning of the task contingencies.

The subsequent reversal phases provided insight into the animals' cognitive flexibility and ability to adapt to changing task demands. Four consecutive reversal phases were implemented, each inverting the previous CS+/CS− contingencies. At the onset of each reversal, a consistent pattern emerged: hit rates decreased significantly, often falling below 20% in the first session, while false alarm rates increased markedly, typically reaching 40–50% (perseverative errors). This increase in false alarms suggests perseveration, a hallmark of cognitive inflexibility during early reversal learning. The d′ values decreased substantially, frequently falling below 0, indicating performance below chance as animals initially perseverate with the previous contingencies.

Following the initial performance decline, animals demonstrated consistent adaptability over the multiple changes of the choice-outcome contingencies. Hit rates gradually increased over subsequent sessions, while false alarm rates steadily decreased. The d′ values recovered, usually reaching criterion levels (d′ > 1) within 3–4 sessions. This pattern of decline

and recovery was observed across all four reversal phases, albeit with some variations in the magnitude and speed of adaptation.

An observed trend was the progressive improvement in reversal learning performance across successive reversals. The magnitude of initial performance decrements tended to decrease with each reversal, and the rate of recovery appeared to accelerate in later reversals. Peak performance levels (both in terms of conditioned response rates and d′ values) showed a general upward trend across reversals. This progressive improvement suggests that animals were developing a more flexible cognitive strategy, allowing for faster adaptation to the changing task contingencies.

Overall, the behavioral data from the multiple reversal Go/NoGo auditory task highlight the gerbils' robust learning dynamics, characterized by initial rapid acquisition of discrimination, followed by adaptive flexibility in response to task reversals. We divided training into three phases: early reversal (d′ < 0, high perseverative errors), intermediate learning (0 < d′ < 1, steepest performance increase), and retrieval (d′ > 1, stabilized performance), enabling analysis of distinct neural patterns. Henceforth, this paradigm is valuable for probing the neural mechanisms underlying cognitive flexibility and auditory-guided decision-making.

**Chronic current source density recordings from the primary auditory cortex**

To investigate the layer-specific cortical activity in the primary auditory cortex (A1) during auditory learning and decision-making, we employed chronic current source density CSD recordings in awake, behaving Mongolian gerbils to investigate the laminar profile of synaptic mass activity on a single-trial basis. The CSD distributions demonstrated the presence of local postsynaptic events from excitatory synaptic populations along the cortical column. Transmembrane currents generated dipoles of sources and sinks, providing a detailed picture of the layer-specific synaptic activity. During the presentation of conditioned stimuli (CS, 200 ms duration), the CSD analysis revealed distinct spatiotemporal patterns of current sinks (blue) and sources (red) across the cortical layers (Fig. 5). These patterns were indicative of the

feedforward flow of sensory information in A1. Notably, in awake recordings, we observed prominent early sinks in the infragranular input layers Vb and to a minor degree, within granular input layers III/IV. Both early sinks reflect the initial thalamocortical input from the medial geniculate body[5,19]. The chronic nature of our recordings enabled us to track changes in these CSD patterns over the course of learning and across different behavioral conditions. This approach provided a unique opportunity to study the layer-specific dynamics of A1 microcircuits during auditory-guided behavior and decision-making processes.

Figure 5a shows representative CSD profiles for hit, miss, false alarm, and correct rejection trials, averaged across reversal sessions. All categories displayed the canonical sequence of early granular and infragranular sinks followed by supragranular activation, but with notable variations in amplitude and temporal persistence. These results demonstrate that laminar CSD activity not only encodes sensory input but also reflects the animals' behavioral choice. To control for potential frequency-specific effects of the two presented sound frequencies, we compared laminar CSD responses to 1 kHz and 4 kHz across reversal sessions, balanced for their roles as CS+ and CS−. Both stimuli evoked the canonical sequence of granular and infragranular sinks followed by supragranular activation, with only minor amplitude differences. We therefore conclude that the laminar effects in Fig. 5a primarily reflect performance-dependent modulation rather than systematic differences between stimulus frequencies.

For further analysis, we focused on the post-CS activity window (500 ms following stimulus onset). This allowed us to examine the sustained and potentially task-related modulation of cortical activity beyond the immediate sensory-evoked response (Fig. 5b). The analysis of the averaged rectified current (AVREC) gives information of the overall temporal current flow across the entire cortical column. We here show the window of the 4th stimulus within each trial, as previous work showed that cortical response differences become most pronounced at this point, when sensory evidence accumulation and behavioral choice converge[5]. A comparison of the RMS-amplitudes within 500 ms windows around the stimulus revealed that as performance improved ($0 < d' < 1$), cortical activity began to differentiate across most behavioral choices, suggesting emerging discriminative abilities. In high performance stages ($d' > 1$), cortical activity was strongest during correct hit trials and significantly different from all other choices, reflecting well-established stimulus discrimination and appropriate behavioral responses. This progression from a generally more undifferentiated overall cortical activity in early, low-performance stages to highly differentiated activity in later, high-performance stages provides insight into the neural mechanisms underlying cognitive flexibility in auditory processing. It demonstrates how A1 adapts to support improved stimulus discrimination and behavioral performance during reversal learning.

## ANOVA analysis of cortical activity before decision making across different layers

To investigate how cortical activity in different layers of A1 relates to behavioral performance and decision-making during reversal learning, we conducted an ANOVA analysis of trace RMS values from CSD traces derived from different cortical layers across different performance levels and behavioral choices. The analysis revealed distinct patterns of cortical activity across layers and performance levels, providing insights into the neural mechanisms underlying auditory processing and learning (Fig. 6).

Whereas Fig. 5 summarizes stimulus-locked responses across the trial sequence[5], we here focused on decision-locked cortical dynamics aligned to the tone immediately preceding the behavioral response, capturing laminar dynamics predictive of the ensuing choice. During low performance stages ($d' < 0$), cortical activity showed minimal differentiation between behavioral choices across most layers. The only significant difference observed was in the infragranular layer Va, which showed slight variance between correct responses (hits and correct rejections) and false alarms, while other layers displayed no significant distinctions between behavioral choices. As performance improved to intermediate levels ($0 < d' < 1$), representing the period of steepest learning, significant differences in cortical activity emerged

across all behavioral choices within multiple layers. The most pronounced differences appeared in the infragranular layers Vb and VI, particularly in distinguishing correct rejections from both false alarms and hits, as well as differentiating between false alarms and misses. Supragranular (I/II) and granular (III/IV) layers also exhibited significant differences between correct rejections and both false alarms and hits, though these differences were less pronounced compared to the deeper layers. This pattern suggests that during active learning, deeper cortical layers play a primary role in representing differential behavioral choices. In high performance stages ($d' > 1$), the pattern of differentiation shifted across cortical layers. The previously prominent differences in infragranular layers became less pronounced, remaining significant only in distinguishing correct rejections from incorrect responses (misses and false alarms). Most notably, the granular layer (III/IV) emerged as the site of strongest differentiation, showing significant differences across all behavioral choices. A similar but less prominent pattern of differentiation was also observed in supragranular layers (I/II). This shift suggests that once the task is well-learned, middle and upper cortical layers become more engaged in discriminating between behavioral choices, while deeper layers maintain a more selective role in error detection.

This progression from minimal differentiation in early, low-performance stages to highly differentiated activity in later, high-performance stages underscores the dynamic nature of cortical processing during learning. The findings suggest that different layers of the A1 process sensory information and behavioral responses differently, with distinct patterns of activity emerging as the animals develop and refine their task-specific processing abilities. We observed that cortical activity patterns changed as learning progressed, with increasing differentiation between behavioral outcomes across layers. While early learning stages showed similar activation patterns across layers, later stages were characterized by distinct layer-specific activity patterns that correlated with behavioral performance. While the causal relationship between these neural changes and behavioral improvement remains to be determined through targeted manipulation studies, these layer-specific changes in neural discrimination provide valuable insights into how the auditory cortex adapts its processing during reversal learning tasks, and hence reflecting neural mechanisms supporting behavioral flexibility.

## Time-resolved GLMM-based effect sizes in auditory cortex layers across different performance levels

To complement our ANOVA analysis of overall activity patterns, we examined the temporal evolution of neural predictors for behavioral choices using time-resolved GLMM-based effect sizes (R2m values). This time-resolved GLMM analysis of layer-specific RMS values around the individual stimuli before a reaction of the animal, revealed both complementary and distinct aspects of layer-specific processing during decision-making (Fig. 7). The figure illustrates the GLMM-based R2m values for different behavioral choices (hit, miss, false alarm, correct rejection) plotted against the number of stimuli before the final decision, after pooling the data across all reversal blocks and splitting them according to the three performance levels based on $d'$.

At low performance levels ($d' < 0$), we observed multiple distinct temporal patterns. Most strikingly, infragranular layer VI showed exceptionally large effect sizes ($R^2m = 0.35$–$0.8$) in predicting hit responses up to 4.5 s before the behavioral choice, which might be consistent with an involvement in early task learning. Additionally, supragranular layers (I/II) contributed significantly to differentiating hits from misses 1.5–3 s before choices. Notably, the comparison between hits and correct rejections engaged multiple layers shortly before decisions, while layer VI showed only moderate effect sizes in distinguishing hits from false alarms - a pattern complementing our ANOVA findings of limited early-stage differentiation. During intermediate performance levels ($0 < d' < 1$), when most learning occurred, the temporal dynamics shifted considerably. While overall effect sizes were somewhat reduced, possibly reflecting increased uncertainty during active learning, we observed earlier onset of choice-predictive activity, extending up to 4.5 s before decisions. Supragranular layers (I/II)

**Fig. 6 | ANOVA analysis of cortical activity before decision making across different layers and performance levels.** Trace RMS values (500 ms window after stimulus onset preceding the reaction time) are plotted with respect to behavioral choices (hit, miss, false alarm, correct rejection) across all reversal blocks (reversals 1–4) and split by performance levels based on the sensitivity index (d'). **Left:** During low performance levels (d' < 0), differences in behavioral choices show minimal significant differences across most layers, except for slight variance in the infragranular Va layer between correct responses and false alarms. **Middle:** At intermediate performance levels (0 < d' < 1), significant differences in cortical activity emerge across all behavioral choices within all layers, particularly in the infragranular layers Vb and VI. **Right:** At high performance levels (d' > 1), the most significant differences in behavioral outcomes are observed, especially in the granular layers (III/IV), indicating robust neural responses and high discrimination ability. Box-plots show median, interquartile range, full data range, and outliers, with significant differences indicated by a two-way rmANOVA with Holm-corrected post-hoc tests. Asterisks indicate statistical significance of differences between groups (*$p \leq 0.05$, **$p \leq 0.01$, ***$p \leq 0.001$, ****$p \leq 0.0001$). See corresponding data for the RMS amplitude of the AVREC waveform in Supplementary Fig. S1.

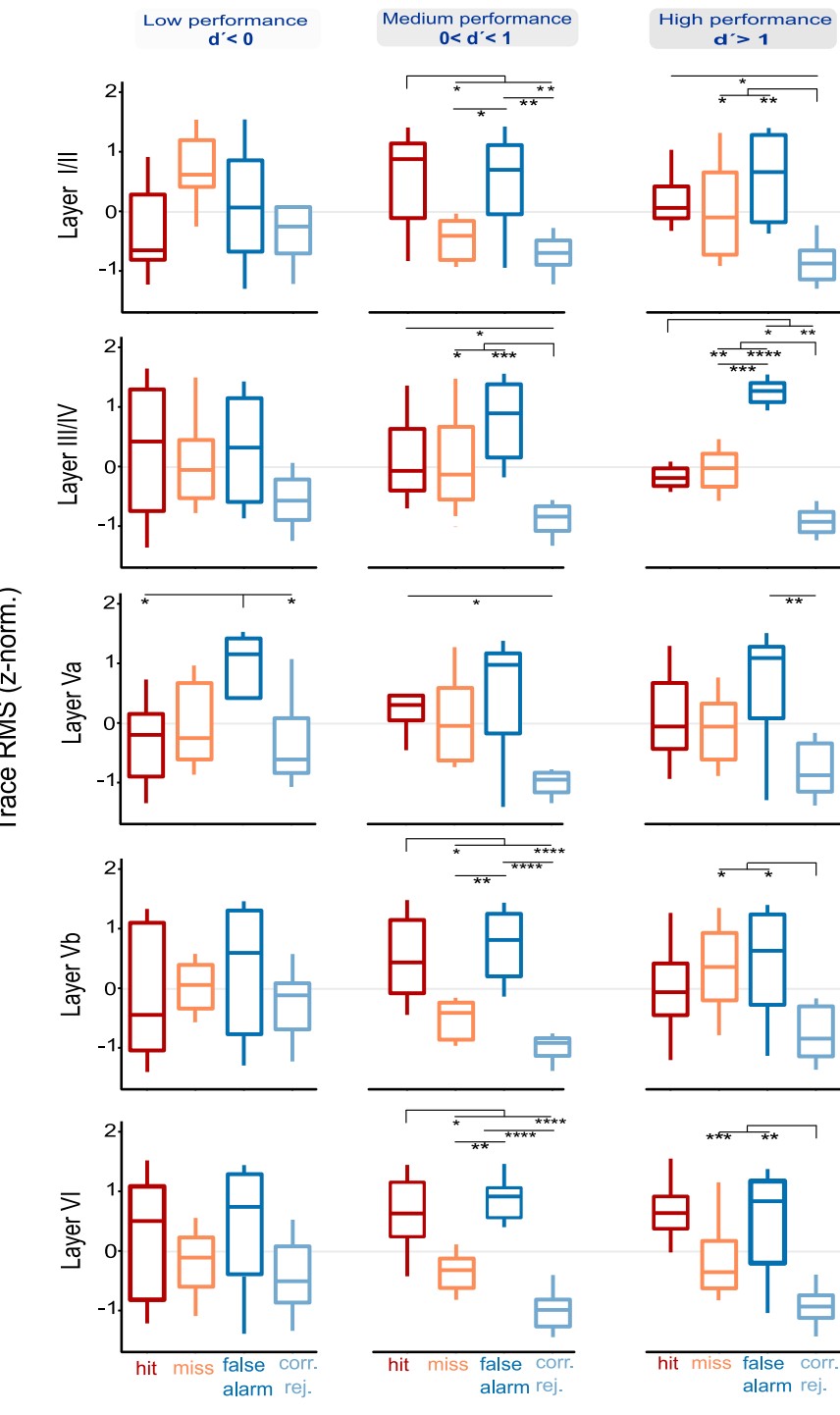

showed large effect sizes in distinguishing hits and false alarms from other behavioral choices. This distributed pattern of activation across the cortical column aligns with our ANOVA results showing increased differentiation during this learning phase, though the GLMM analysis revealed additional temporal complexity in how these distinctions evolved. At high performance levels (d' > 1), the temporal dynamics of choice prediction became more refined. Layer VI maintained strong effect sizes for hit responses versus misses and correct rejections, a pattern now shared by granular layer III/IV, though primarily concentrated just before choice execution. Correct rejections versus false alarms showed extended predictive power in layer III/IV up to 3 s before choices. Importantly, high performance states revealed new distinctions between incorrect responses (false alarms vs. misses) primarily in deeper layers, while misses versus correct rejections showed

increasing differentiation with longer accumulation times (3–4.5 s) in layers I/II and III/IV. While these patterns partially align with our ANOVA findings of increased granular layer involvement during high performance, the GLMM analysis revealed distinct temporal sequences in how these choice-predictive signals evolved.

This time-resolved analysis extends the ANOVA results by revealing how choice-predictive neural signals develop over time across cortical layers. While both analyses highlight the dynamic nature of cortical processing during learning, the GLMM approach specifically shows how different layers contribute to evidence accumulation before choices are made, with patterns varying markedly across performance levels. These complementary analytical approaches together provide a more complete picture of how cortical circuits adapt during reversal learning.

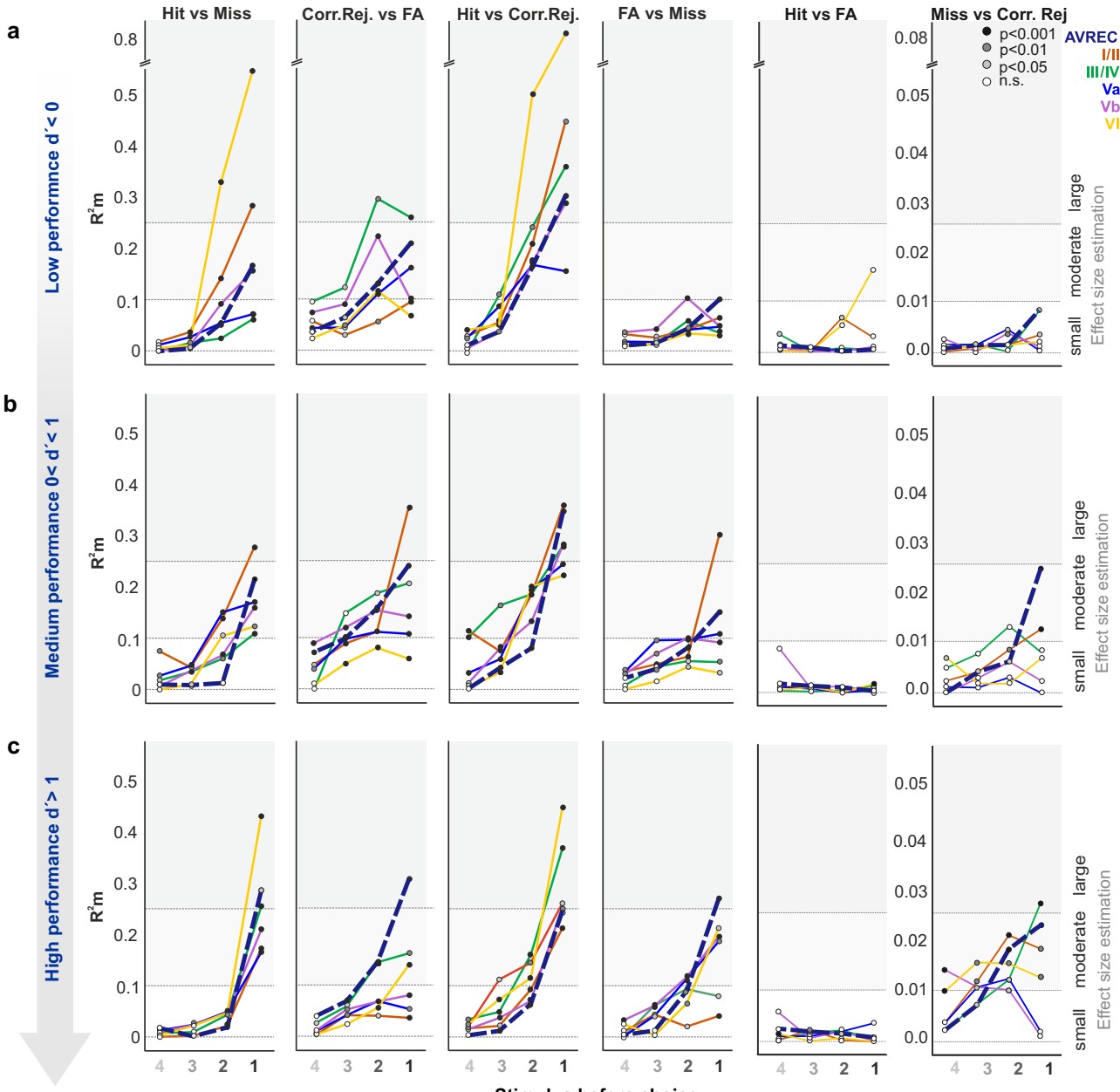

**Fig. 7 | Time-resolved GLMM-based effect sizes of behavioral outcomes at three different performance stages during multiple reversal training.** GLMM-based R2m values for different behavioral choices (hit, miss, false alarm, correct rejection) plotted against the number of stimuli before the final decision, after pooling the data across all reversal blocks and splitting them according to three performance levels based on d'. **a** At low performance levels (d' < 0), infragranular layer VI shows extremely large R2m values for predicting hit responses, indicating its crucial role in early learning stages. **b** At intermediate performance levels (0 < d' < 1), a more balanced distribution of effect sizes across layers is observed, with supragranular layers I/II showing the highest predictive power for correct rejections. **c** At high

performance levels (d' > 1), large R2m values in layer VI persist, though they are smaller than at low performance levels, suggesting a more distributed neural processing as performance improves. Across all levels, strong effect sizes for hit versus miss comparisons are noted, particularly in infragranular layers, while moderate to strong effect sizes for correct rejections versus false alarms are seen, especially at intermediate levels. This figure underscores the dynamic role of cortical layers in auditory processing and decision-making during learning. Statistical significance of differences between groups ($p \leq 0.05$, $p \leq 0.01$, $p \leq 0.001$) is indicated by the color of the data points.

## Spectral power analysis during multiple reversal tasks

The GLMM analysis identified significant temporal dynamics of cortical activity across behavioral performance levels, revealing distinct neural processing patterns in response to correct and incorrect choices. To further characterize these findings, we conducted a spectral power analysis (Fig. 8). We here focused on behavioral choices that were most pronounced in the GLMM analysis (Fig. 7) with respect to error-processing mechanisms (Figs. 9 and 10) and neural signatures associated with motor initiation (hits)

and response inhibition (correct rejections), providing a framework to examine sensory-motor decision processes (Fig. 11). Given that main effects were found across supragranular, granular, and infragranular layers, we analysed layers I/II, III/IV and VI, each of which plays a distinct role in cortical processing: Layer I/II is related to cortico-cortical communication and implicated in cognitive control mechanisms, layer IV serves as the primary recipient of thalamocortical sensory input, and layer VI is associated with feedback processing and sensory-motor integration[35].

**Fig. 8 | Spectral power analysis during multiple reversal tasks between hits and miss trials.** This figure presents the spectral power computed using CWT to obtain time-frequency representations of the CSD signals during multiple reversal tasks, comparing hits and miss trials. **a** Magnitude scalogram of hit trials showing wavelet power across time and frequency for all subjects (*n* = 8), averaged over multiple sessions and trials. **b** Magnitude scalogram of miss trials showing wavelet power across time and frequency for the same subjects using identical CWT parameters. **c** Difference scalogram between hit and miss trials, illustrating differences in spectral power. Clusters observed in the actual group comparison that exceeded 95% of the sizes found in the per-mutation distribution were deemed significant and are plotted by white boundaries. **d** Effect size matrix (Cohen's d) indicating the strength of differences in spectral power across time and frequency between hit and miss trials. This analytical approach reveals significant differences in neural oscillatory activity between behavioral outcomes.

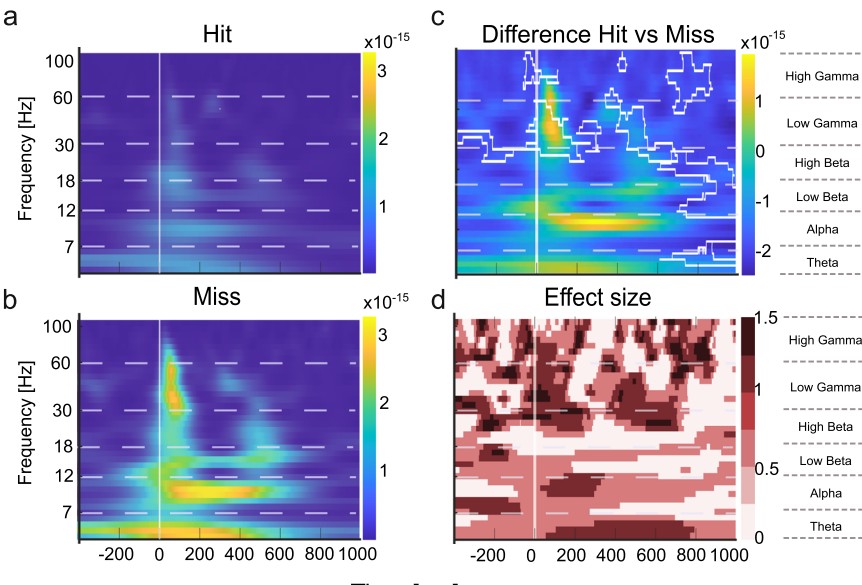

Using time-frequency decomposition via continuous wavelet transform (CWT), we quantified the evolution of frequency-specific neural activity across these layers, tracking how oscillatory dynamics relate to decision-making processes as learning progresses. Unlike time-domain analyses, which only reveal broad changes in signal amplitude, spectral power analysis offers a frequency-domain perspective, allowing us to examine how different frequency bands contribute to behavioral responses. We performed the analysis using the Continuous Wavelet Transform (CWT), which converts time-domain signals into time-frequency representations (scalograms) reflecting both amplitude and phase information. This method captures dynamic changes in neural oscillations associated with different behavioral outcomes, providing detailed insights into frequency-specific activity driving these outcomes. We focused on key frequency bands associated with various cognitive and sensory processes: theta (4–7 Hz), alpha (8–12 Hz), low beta (13–18 Hz), high beta (19–30 Hz), low gamma (31–60 Hz), and high gamma (61–100 Hz).

To quantify differences in spectral power, Fig. 8 shows exemplarily how neuronal activity during hits and miss responses contrast (Fig. 8a, b). We employed permutation cluster-mass analysis (Fig. 8c), where warmer colors indicate higher power for miss trials relative to hits and corresponding effect size calculations (Cohen's d; Fig. 8d). In panel c, significant clusters (at least 5 × 5 pixels) that exceeded 95% of the sizes found in the permutation distribution, and showed Cohen's d-effect sizes of >0.4, are indicated by white boundaries. Complementing this statistical map, panel (d) displays the effect size distribution (Cohen's d) across the full time–frequency matrix, illustrating the magnitude of hit–miss differences independently of statistical thresholding. This highlights consistent differences in beta- and gamma-band activity aligned to stimulus onset, particularly in miss trials. Figure 8 illustrates how spectral analysis captures oscillatory differences between hit and miss trials, with significant clusters emerging in the beta- and gamma-bands. This example demonstrates the sensitivity of the approach and provides the framework for the subsequent layer-resolved analyses in Figs. 9–11.

In detail, we found layer- and frequency-specific modulations in response to correct (Hit) and incorrect (Miss) choices, evolving systematically across performance levels (Fig. 9). During the low-performing state (d' < 0, reversal phase), layers I/II showed background gamma activity, while layers IV and VI exhibited minimal spectral differences, indicating underdeveloped task-specific processing. At the intermediate learning phase (0 < d' < 1), stimulus-coupled beta and gamma activity emerged in layers VI. At high performance (d' > 1,

retrieval phase), particularly layers I/II and III/IV showed high gamma power during stimulus presentation, which might be related to decision execution. Layer III/IV also maintained broad beta activity across all time periods during high performing states, which might indicate a robust sensory representation. In contrast, during the early reversal phase, significant differences in gamma activity were only found in layers I/II and were more spatially and temporally diffuse.

When comparing false alarms and correct rejections (Fig. 10), particularly layers I/II showed distinct cortical dynamics across performance levels. During the low-performing state (d' < 0, reversal phase), specifically layers I/II exhibited broadband power increases from alpha to gamma bands at stimulus onset. During the learning phase (0 < d' < 1, learning phase), there are only some higher-gamma background differences across layers. In the high-performance state (d' > 1, retrieval phase), layers I/II and IV exhibited broadband power increases throughout the full background, aligning with anticipatory and feedback-related processing, while layer VI mainly displayed background high gamma activity, reinforcing its role in error processing and response adaptation.

Spectral power differences between hits and correct rejections (Fig. 11) varied across performance levels. During the low-performing state (d' < 0, reversal phase), no strong spectral differences were observed. In the intermediate learning phase (0 < d' < 1), layer VI showed stimulus-locked high-beta and gamma activity, followed by post-stimulus low-beta power. In the high-performance state (d' > 1), significant post-stimulus differences were confined to layer III/IV, spanning low-beta to gamma frequencies, while layers I/II and VI did not show notable effects.

Collectively, our spectral power analysis revealed how error processing and decision-making evolve across cortical layers and frequency bands throughout learning. During the reversal phase (d' < 0), spectral activity in layer I/II was broadly distributed, particularly in the gamma and beta bands, in line with diffuse cortical engagement and task adaptation. In the learning phase (0 < d' < 1), stimulus processing became more structured, with gamma-band activity emerging in both supragranular and infragranular layers, supporting sensory-motor integration and decision refinement. Finally, in the retrieval phase (d' > 1), background beta and gamma signatures were most pronounced in supragranular and granular layers, stabilizing learned sensory representations and response selection. These findings highlight a hierarchical shift from broad, unspecific cortical engagement in early learning to precise stimulus processing during task acquisition and, ultimately, to anticipatory and feedback-related processing in well-learned states.

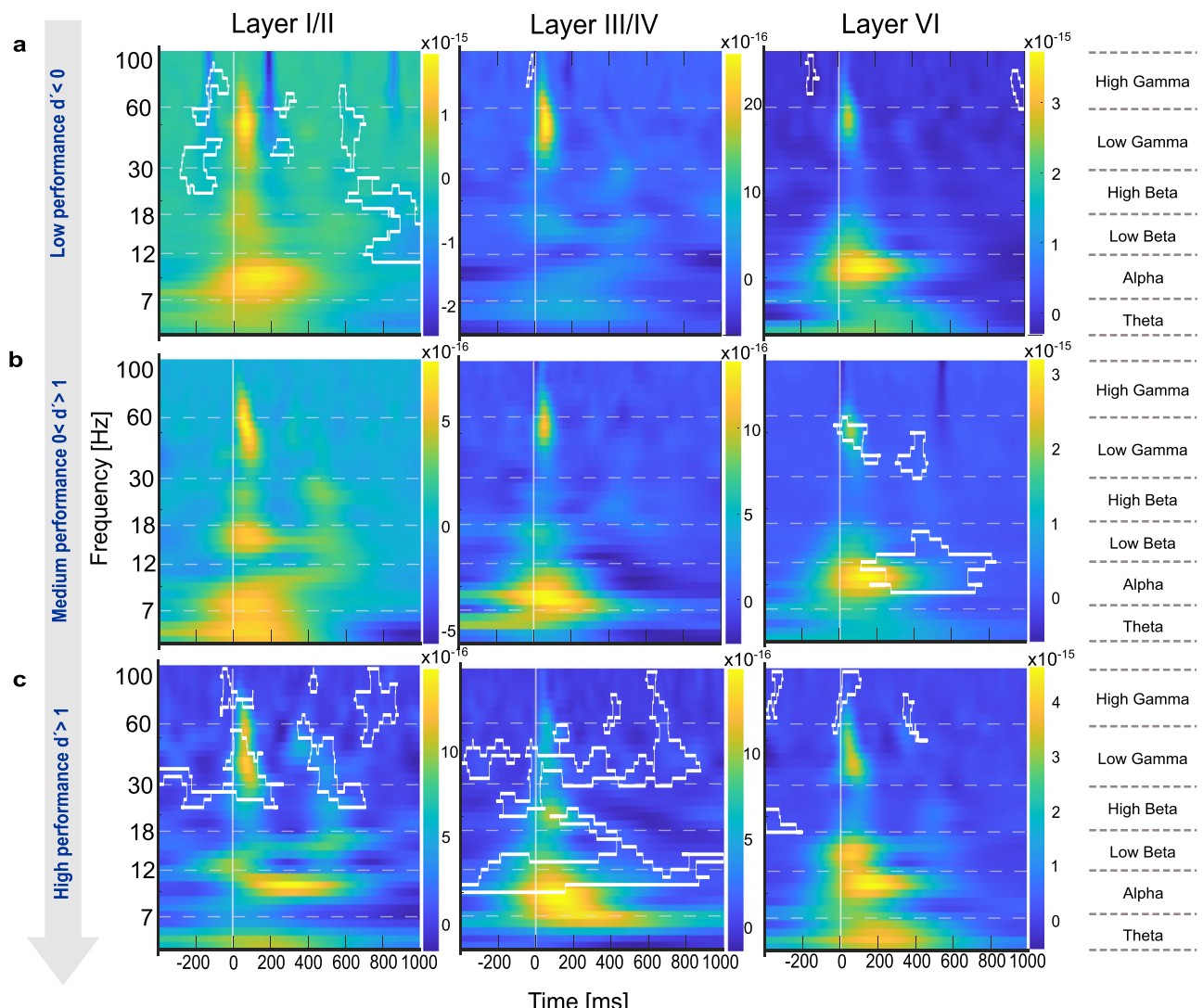

**Fig. 9 | Spectral power differences for Hit vs. Miss comparisons across cortical layers and performance levels.** Time-frequency plots show spectral power changes in layers I-II, IV, and VI for hit vs. miss contrasts across the three performance levels (d' < 0, 0 < d' < 1, d' > 1; **a–c**). Significant clusters are outlined in white ($p < 0.05$). Layers I/II and III/IV showed sustained gamma power during stimulus periods in high-performance states. Layers III/IV also showed broad beta power across the entire time window. Layer VI showed significant differences in stimulus-coupled beta and gamma power only during the medium performance level.

## Discussion

The primary auditory cortex (A1) plays a crucial role in integrating sensory representations with task-related decision-making processes by encoding both stimulus properties and behaviorally relevant task contingencies[3–6]. In this study, we examined layer-specific temporal and spectral dynamics of auditory decision-making in Mongolian gerbils using chronic laminar CSD recordings during a multiple reversal learning task. As discrimination performance improved and animals adapted to contingency reversals, CSD recordings revealed systematic changes in laminar activity patterns across performance levels. Early in learning, deep-layer activity predominated, whereas with training, engagement of supragranular layers (I/II) increased. During active learning, stimulus-locked beta- and gamma-band oscillations in deeper layers reflected enhanced sensory-motor integration. In the retrieval phase, supragranular and granular layers (I/II and III/IV) showed the strongest choice-related differentiation, with background beta and gamma oscillations that would be consistent with stabilized sensory representations and refined decision-making. These findings highlight an adaptive reorganization of A1 circuits that supports cognitive flexibility during auditory decision-making.

### Behavioral and neural adaptation during reversal learning

The multiple-reversal Go/NoGo task revealed distinct phases of auditory learning and cognitive flexibility (Fig. 4). During initial discrimination, hit rates rose toward ~80% while false alarms stabilized near ~20%, yielding a steady increase in d'. At each reversal, performance transiently deteriorated (hits <20%, false alarms 40–50%, d' < 0), consistent with perseverative responding and proactive interference. Performance typically recovered within 5–6 sessions, and the depth of the reversal-induced performance drop diminished across successive reversals, indicating improved rule-switching and feedback-guided adaptation[8,13]. Importantly, the observed increase in perseverative errors immediately after reversals suggests that animals did not anticipate rule changes based on the fixed reversal interval. This argues against the notion that a global reversal-prediction rule was learned and instead supports a model of reactive, feedback-driven adaptation.

CSD recordings paralleled these behavioral phases with distinct laminar signatures (Figs. 6–11). Early after reversals, modulation during hit trials was strongest in layers I/II and VI (Fig. 7). Supragranular I/II showed shifts in background activity, while layer VI exhibited enhanced response-related activity; both patterns are consistent with task updating, feedback processing, and motor engagement[6,7,16,35–39]. As learning progressed, deep-

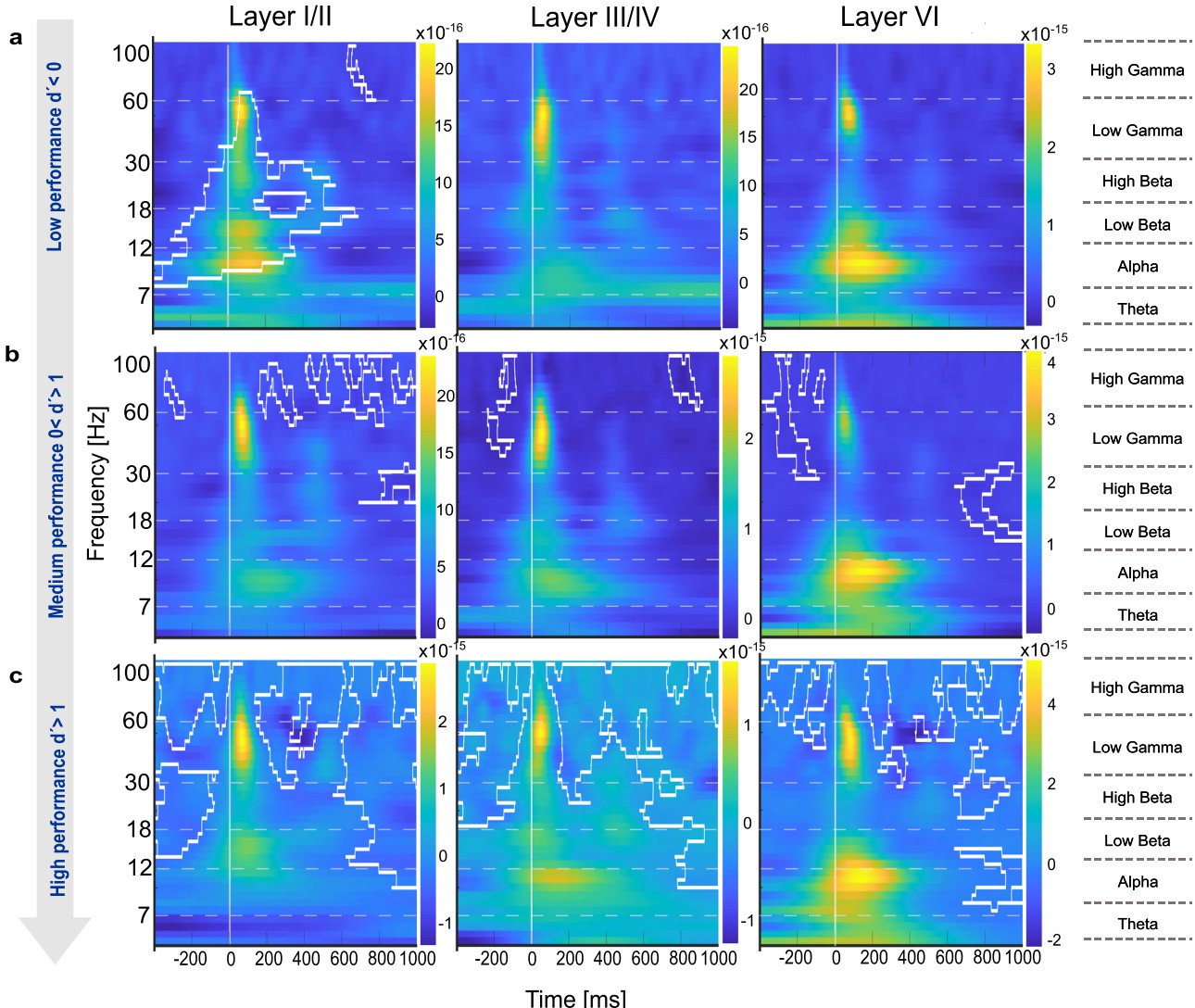

**Fig. 10 | Spectral power differences for FA vs. Corr. Rej. comparisons across cortical layers and performance levels.** Spectral power dynamics are depicted for layers I-II, IV, and VI in the FA vs. Corr. Rej. contrast across performance levels (d' < 0, 0 < d' < 1, d' > 1, **a–c**). Statistically significant clusters ($p < 0.05$) are marked with white outlines. Layers I/II show broad power differences in the low performing state (d' < 0). At the intermediate performance level (0 < d' < 1), high-beta and low-gamma activity was present across layers. In the high-performance state (d' > 1), layers I/II showed broadband increases extending before and after stimulus presentation, layer III/IV displayed sustained beta- and gamma-band activity during the stimulus period, and layer VI showed stimulus-coupled beta/gamma together with enhanced background high-gamma power.

layer prominence gave way to broader recruitment. At intermediate performance, VI showed stimulus-locked buildup compatible with accumulating-evidence dynamics (Fig. 7).

In the high-performance state, layer IV was most engaged during response selection (Fig. 6), coinciding with enhanced beta-band activity (Fig. 11). Beta oscillations have been associated with the stabilization of learned stimulus–response contingencies and a reduced reliance on exploratory strategies[40,41]. Layer VI continued to show large effect sizes in hit contrasts (Fig. 7), consistent with a role in error feedback via corticothalamic and corticocortical loops[42]. Deep-layer outputs have been shown to transmit motor-related signals to subcortical targets such as the striatum and superior colliculus[38], supporting their contribution to adaptive response selection in dynamic task environments. High decision accuracy was accompanied by more structured coordination across cortical layers, consistent with the model of Tovar et al.[43], in which precise laminar interactions optimize stimulus evaluation and response selection[43].

Taken together, these results point to a phase-dependent redistribution of processing in A1: deep layers initially accompany trial-and-error adaptation by integrating sensory input with feedback, whereas

refined stimulus–response mapping during proficient performance relies more on granular and supragranular circuits (Figs. 9–11; summary in Fig. 12). This progression aligns with frameworks in which rodents employ both exploratory (model-free) and inference-based strategies during reversal learning[8,44,45], even though our task does not distinguish these directly. Overall, the data support a hierarchical organization in A1 whereby early adaptation coincides with infragranular feedback engagement and later retrieval features coordinated activity in granular and supragranular layers[3,5,6].

### Layer-specific spectral dynamics during decision-making

While stimulus-evoked synaptic responses (Figs. 6 and 7) provided insights into localized current flow and net synaptic input strength, spectral power analysis (Figs. 8–11) revealed broader network-level coordination patterns and frequency-specific neural dynamics that evolved across learning phases Supragranular layers (I/II) exhibited distinct spectral modulations across different task phases, suggesting a progressive role in sensory processing, task reconfiguration, and top-down response modulation. Although their overall activity was weaker and less differentiated compared to deeper layers,

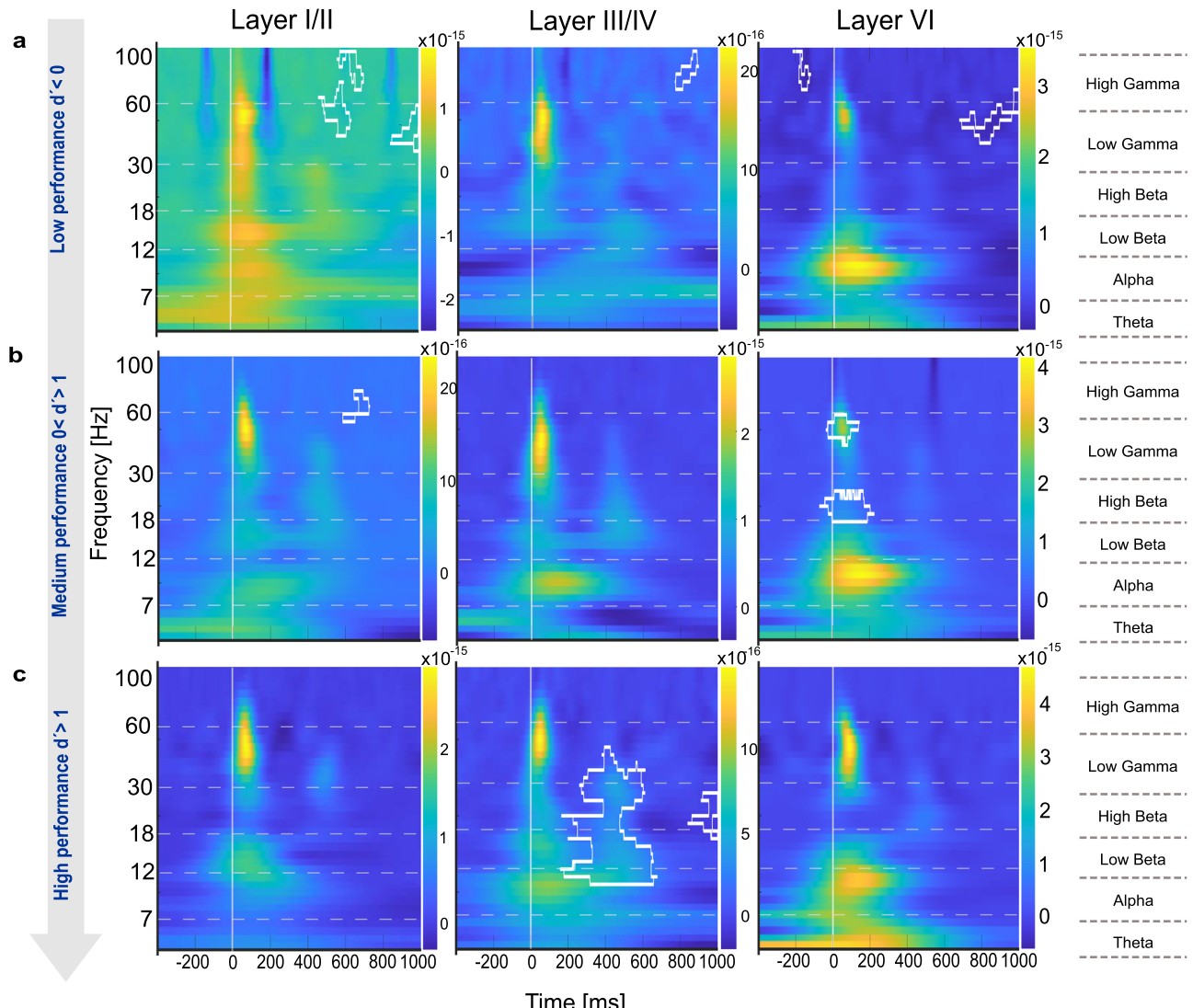

**Fig. 11 | Spectral power differences for Hit vs. Corr. Rej. comparisons across cortical layers and performance levels.** Spectral power dynamics are depicted for layers I-II, IV, and VI in the hit vs. corr.rej. contrast across performance levels (d' < 0, 0 < d' < 1, d' > 1, **a–c**). Significant clusters ($p < 0.05$) are indicated by white outlines. The main effect was observed in the high-performance state (d' > 1), where layer III/IV exhibited a broad alpha- to beta-band power increase shortly after stimulus onset.

supragranular circuits likely contributed to early sensory processing and top-down feedback mechanisms during stable task performance[2,36,46]. During early reversal learning (d' < 0), supragranular layers I/II showed significant beta- and gamma-band differences, particularly in the background (Figs. 9–11). This pattern likely reflects early sensory evidence accumulation and preparatory engagement in adaptive processing mechanisms. This layer-specific analytic framework provides a measure of how dominant activity progressively shifts from deep to superficial cortical circuits, reflecting the laminar reorganization that accompanies behavioral transitions.

As learning progressed, spectral power differences became increasingly localized, with stimulus-coupled gamma oscillations emerging in layers I/II and VI (Fig. 10), consistent with a heightened recruitment of cortico-cortical processing networks. In support, Deliano et al.[47] showed that dopaminergic neuromodulation enhances high-gamma phase-locking in A1, particularly in supragranular layers III/IV, indicating neuromodulatory shaping of gamma-band responses under behaviorally relevant conditions. This is in line with the idea that gamma oscillations facilitate cortical communication and cognitive flexibility, allowing for dynamic task adaptation by coordinating activity across neural populations[48–50]. In our data, stimulus-locked gamma oscillations in layer VI (Figs. 9–11) further support its involvement

in integrating sensory and motor signals. During retrieval, broadband beta-to-gamma transitions became prominent both before and after stimulus presentation, particularly during error-related processing in layers I/II and III/IV (Hit vs. Miss, FA vs. CR; Figs. 9 and 10). As learning progressed, differences in spectral power between correct and incorrect choices became more pronounced, consistent with the increasing engagement of distributed cortical circuits in auditory-guided decision-making[36,51]. These oscillatory signatures suggest that beta-gamma interactions play a crucial role in error-related performance adjustments, enhancing stimulus-response consolidation through adaptive cortical reorganization.

Layer VI showed both background and stimulus-coupled high-gamma activity during error processing in the retrieval phase (Figs. 9, 10), underscoring its role in sensory-motor integration. However, while VI was strongly engaged during early and intermediate learning (Figs. 9–11), its oscillatory involvement diminished during correct choices in the high-performance state (Fig. 11). This pattern suggests that VI contributes primarily to decision refinement during learning, while direct task execution becomes more automated and less dependent on infragranular activity once contingencies stabilize.

Finally, during correct response initiation and inhibition (Hit vs. CR; Fig. 11), spectral differences were confined to a post-stimulus low-beta to

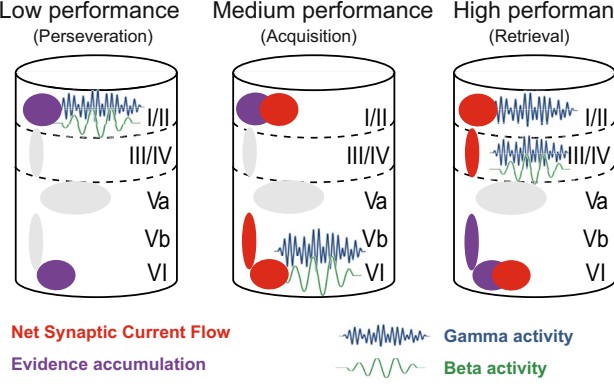

**Fig. 12 | Summary schematic of laminar dynamics across behavioral performance phases.** Schematic overview integrating findings from net synaptic current flow (Fig. 6), evidence accumulation based on GLMM analyses (Fig. 7), and spectro-temporal contrasts (Figs. 9–11). Each cortical column illustration depicts the predominant laminar effects observed during (left) low performance/perseveration (d′ < 0), (middle) medium performance/acquisition (0 ≤ d′ < 1), and (right) high performance/retrieval (d′ > 1). CSD amplitudes reflect the strength of net synaptic current flow dominated by excitatory postsynaptic currents indicated here by red shapes (CSD-based RMS); purple shapes indicate layers with significant evidence accumulation (GLMM effect sizes). Oscillatory components (blue = gamma, green = beta) are shown only where significant differences were observed in the spectro-temporal analyses (Figs. 9–11). Key findings include: (i) diffuse recruitment of I/II and VI during low performance, (ii) increasing involvement of I/II and VI during acquisition, and (iii) more focal signatures at high performance, with gamma in I/II (Hit vs Miss, Fig. 9) and beta in IV (Hit vs Correct Rejection, Fig. 11) emerging as characteristic markers of stabilized discrimination. This schematic highlights the transition from broadly distributed laminar engagement to refined, layer-specific signatures of adaptive decision-making.

gamma shift in layer IV, consistent with its established role as the main site of thalamocortical input[18,21]. The emergence of stimulus-locked gamma oscillations in layers I/II and VI likely reflects a shift from broad exploratory activation toward more structured cortical processing[47,50], optimizing efficiency as performance stabilizes. This progression illustrates a transition from early, diffuse cortical engagement to later refinement within well-defined sensory–motor circuits.

### Dynamic cortical functional reorganization underlying adaptive learning

This study provides novel insights into how the primary auditory cortex dynamically reorganizes its synaptic and spectral activity across different stages of learning and decision-making. The progressive refinement of stimulus-response associations was reflected in a shift from deep-layer dominance during early adaptation to more structured engagement of granular and supragranular circuits as task proficiency increased. Our findings highlight the essential role of dynamic cortical plasticity in supporting cognitive flexibility, with early-phase broad encoding giving way to specialized processing as learning stabilizes. This aligns with previous work demonstrating that A1 exhibits a hierarchical organization, in which deep layers are crucial for initial stimulus integration and adaptation, whereas supragranular circuits play a dominant role in refined sensory representations and task execution[5]. The spectral dynamics observed in supragranular and infragranular layers underscore their complementary roles in integrating sensory input with behavioral adaptation, contributing to task rule updating, response selection, and error monitoring, particularly layer Vb, exhibit frequency-specific modulation in response to contextual or neuromodulatory input[52]. These results further support the layer-specific specialization of A1 in flexible decision-making and reinforce the importance of spectral signatures in shaping adaptive neural processes. By demonstrating how learning-related changes in cortical activity correspond to distinct behavioral states, this study advances our understanding of the neural basis of sensory

decision-making and provides a framework for investigating how distributed cortical circuits support cognitive flexibility in changing environments.

While our study focused on laminar processing within the auditory cortex (A1), future research may benefit from examining how A1 interacts with higher-order cortical areas involved in behavioral flexibility. The prelimbic cortex, a subregion of the medial prefrontal cortex (mPFC), has been shown to modulate sensory cortex activity and is implicated in cognitive control and rule updating. Recent work has highlighted coherence between the prelimbic cortex and motor regions during auditory-guided decision-making[53], and inactivation of prefrontal areas has been shown to reduce context sensitivity in visual cortex[54]. In addition, neuromodulatory systems such as dopaminergic input to auditory cortex can influence gamma-band synchrony and phase-locking, and may contribute to the dynamic regulation of cortical responsiveness during learning. These findings suggest that cross-regional dynamics, including prefrontal feedback, may complement local plasticity mechanisms in shaping adaptive auditory behavior.

## Data availability
Preprocessed data are available via the following link: https://ncloud.lin-magdeburg.de/s/Dt4EEY5WENrzo7M. Raw data (plx-files and converted Matlab-files) are available upon request from the corresponding author.

## Code availability
Data analysis code is available via the following GitHub repository: https://github.com/CortXplorer/Acun_et-al.

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

## Acknowledgements

The study was funded by the Deutsche Forschungsgemeinschaft to MH (Project-IDs: 532212224 and 425899996e). We would like to thank Kathrin Ohl for technical assistance. The authors acknowledge the use of AI-based tools (ChatGPT, Perplexity.ai, and Elicit) for assistance in idea generation, conducting literature searches, and optimizing the text. The authors critically reviewed and finalized all content to ensure its accuracy, originality, and

scientific integrity. The authors declare no competing interests. Correspondence and requests for materials should be addressed to M.H.

## Author contributions
M.F.K.H. and F.W.O. designed research and conceived the study. M.Z. collected experimental data. E.A., M.Z., V.K., and K.E.D. analyzed data. M.F.K.H. and M.D. supervised experiments and data analysis. M.D. established the setup and stimulus protocols. M.F.K.H. wrote the manuscript. All authors discussed and reviewed the manuscript.

## Funding

## Competing interests
The authors declare that they have no competing interests, either financial or non-financial, that could have influenced the design, analysis, or interpretation of this study.
