## [Transparent Peer Review file · Communications Biology]

Layer-Specific Cortical Dynamics During Transitions from Error Monitoring to Decision Execution in Reversal Learning

Corresponding Author: Professor Max Happel

Version 0:

Reviewer comments:

Reviewer #1

(Remarks to the Author)

The work entitled "Cortical Dynamics Controlled by Deep and Superficial Layers During Transition from Error Monitoring to Decision Execution in Reversal Learning" addresses an important gap in current knowledge. The authors demonstrate a hierarchically structured response across A1 layers. They use a reversal learning paradigm, switching the rule required to obtain a US (GO/no-GO). The results show that layer VI is mainly involved in the early stages of the learning process, while middle and upper layers are more engaged once learning is well established. The authors also perform spectral analysis of the data, revealing a dynamic organization of A1 responses, with beta and gamma bands predominantly active in deep and upper layers during the active learning phase. However, once learning is established, both bands are predominantly active in the middle and upper layers.

The work is well written, the figures are illustrative, and the results are well connected to previous literature. The experimental design is clear and elegant. However, some comments are described below.

Page 10: "Each daily session contained 60 trials (30 CS+, 30 CS-)". I suggest including this information also in the Methods section to improve clarity and reproducibility.

Page 19: "When comparing false alarms and correct rejections (Figure 10), layers I/I showed distinct cortical dynamics across..." appears to contain a typographical error. It should read "layers I/II" instead of "layers I/I."

Figure 2: It would be helpful to clarify what "RT" stands for in the figure. Additionally, it is not clear how long each trial lasts; specifying the trial duration would improve understanding.

Figure 4:

- Please check whether the X-axis is correctly placed. It appears displaced (e.g., 0 is shown after the first bar).
- In the figure legend, blue and red circles are mentioned, but they do not appear in the figure.
- The sentence "Vertical dashed lines indicate..." seems inconsistent with the figure, as no vertical dashed lines between reversal sessions are visible. It seems the figure was modified after the legend was written. Please review the figure and update the legend accordingly.
- As a suggestion, it might be helpful to include a horizontal shaded frame for each performance criterion (low, medium, and high), similar to the shading previously used for sessions. This could enhance the readability of the figure.

Figure 5: The colors of the box plots appear different from those shown in the legend, making the figure harder to interpret. Please adjust the legend colors to match the box plots.

Figure 5 Legend: The statement "CSD sink activity differs significantly from early stages..." seems inappropriate, as it implies statistical significance based on representative averaged profiles. Please clarify whether this difference is statistically tested, and if not, adjust the wording to avoid suggesting statistical significance.

Figure 6: The results are beautiful, and the color legend matches the box plots correctly. However, it seems that part of the figure might be missing. The figure legend mentions "Top: Trace RMS values (500 ms window before reaction time) plotted with respect to behavioral choices..." but this corresponding part is not visible in the figure. Please review and ensure the figure matches the description in the legend.

Figure 7:

It is difficult to distinguish which line corresponds to each layer based on the current color scheme. I suggest the authors improve the color contrast and perhaps vary the line widths to enhance readability.

Figure 8:

The figure appears to be missing the labels (a), (b), and (c) referenced in the figure legend. Additionally, where (e) is indicated, it may actually correspond to (d). Please revise the labeling to match the legend description accurately.

I consider it interesting for future work to study the role of the medial prefrontal cortex (mPFC) by performing comparative analyses with the current findings. Since A1 receives inputs from other cortical areas (Sakata and Harris, 2009), including the motor cortex (Nelson et al., 2013) and the mPFC—areas especially implicated in behavioral responses and cognitive flexibility—it would be valuable to explore their contribution further. Interestingly, a recent study using a GO/no-GO task, which recorded from several implicated areas, found strong coherence between the prefrontal cortex and other regions, such as the primary motor cortex, (Muñoz-Redondo et al., 2024). Previous studies in visual cortex (V1) have shown that deactivation of the PFC reduces responsiveness to contextual information (Hamm et al., 2021).

Reviewer #2

(Remarks to the Author)

In this paper, Zempeltzi et al. report on the relationship between dynamic changes in neuronal oscillatory activity across layers of the primary auditory cortex and adaptive behavior during reversal learning tasks in Mongolian gerbils. Behavioral experiments involving multiple reversal learning tasks revealed that the animals rapidly adapted to changes in task rules and showed significant improvement in auditory discrimination over several weeks of training. Simultaneously, the authors chronically recorded local field potentials across cortical layers and applied current source density (CSD) analysis during the behavioral tasks. The CSD data, along with subsequent spectro-temporal analyses, showed that both the CSD profiles and specific frequency bands of oscillatory activity changed in a layer-specific and learning-phase-dependent manner. These dynamic, layer-specific changes in neuronal oscillations across different learning phases are potentially interesting and provide valuable insights for readers of the journal. However, there are several concerns regarding the authors' analysis and interpretation of the results, which should be addressed and clarified.

Major Concerns

1: There are too many interpretative statements throughout the manuscript — in the title, abstract, and results sections — that are not directly supported by the observations in this study or by solid evidence from previous research. For example, the title "Cortical Dynamics 'Controlled' by Deep and Superficial Layers..." implies some form of experimental manipulation of activity in the superficial and deep layers, leading to a causal effect on overall cortical dynamics. Readers would reasonably expect such an intervention based on the wording. Another example appears in the abstract, line 10: "In the learning phase, stimulus-locked beta and gamma oscillations in upper and deeper layers 'signified sensory-motor integration'." This is just one of many possible interpretations, but it is difficult to accept without experimental or analytical manipulation of relevant sensory-motor circuits to test the effects on these oscillations. Additional examples are noted in the minor comments below. These kinds of interpretations, which go far beyond the actual observations, risk misleading readers. Interpretations should be made with caution, primarily should be presented in the Discussion section, and, where included elsewhere, should be clearly grounded in solid prior findings.

2: 1) Although the layer-specific changes in oscillatory activity across different learning phases provide valuable information, the overall analyses are not well integrated, making it difficult for readers to grasp the key findings. Summary analyses or schematic overviews should be added to help readers better understand the essential points conveyed by the individual figures — particularly those following Figure 7. Figure 7 presents 18 panels, each showing time-dependent changes in GLMM-based effect sizes overlaid across six different layers and measures. It is challenging to assess the differences between individual lines (representing different layers) across varying behavioral performances and choices. Figures 8 through 11 contain approximately 30 panels of CWT plots under various conditions, showing modest or subtle changes in some cases. The large number of similar panels may make it difficult for readers to follow the findings.

2): Additionally, the authors applied a non-parametric permutation test to identify significant differences between groups of behavioral choices for each condition. However, given that the statistical testing spans five experimental dimensions — (1) behavioral performance level, (2) cortical layer, (3) oscillation frequency band, (4) time from stimulus onset, and (5) behavioral choice combinations — multiple comparison issues should be addressed. At a minimum, statistical significance should be corrected for multiple comparisons involving at least two factors: behavioral performance levels and cortical layers, particularly in Figures 9, 10, and 11.

3: Regarding the spectro-temporal analysis of layer-specific oscillatory activity, as described under "Spectral Power Analysis" in the Methods section, the continuous wavelet transform was applied to CSD signals. However, it is unclear why the wavelet transform was applied to the spatial second derivative signals (CSD) along the cortical depth axis, rather than directly to the local field potentials (LFPs) at each recording site. This approach may distort the original frequency content of the field potentials due to the additional spatial smoothing inherent in CSD computation. The authors should clarify the rationale for this choice; otherwise, the spectral analysis should be performed directly on the LFP signals.

Minor Concerns

- 1: Line 14 in abstract section, it is unclear why these findings and the main story of the study could be specifically relevant for the research of neurodegenerative diseases.
- 2: On Page 4, in the paragraph of "Behavioral Training Protocol", what kind of criteria was applied for the successful acquisition?
- 3: In Figure 2b, what is RT? What does the difference between the timing of lightening mark and the yellow box of US mean? The length of the time interval of 14-26 s looks shorter than the initial 6 s.
- 4: For the sentence of "... This preprocessing step was crucial to ensure data quality..." in line 32 in the section of Current Source Density (CSD) Analysis on Page 6, what is the quantitative criteria for this preprocessing step?
- 5: In the section of CSD parameter analysis, since AVREC was originally defined in datasets under anesthesia (Givre et al., 1994; Schroeder, 1998), it might be a bit unclear whether to use the same criteria (not only the way of rectification but also the time window) under awake recording condition is valid.
- 6: In Figure 3 caption, where is the "red frame"?
- 7: In line 1 on Page 8, these random variables should be described together with the formula above in Page 7 to help readers to understand in consistent manner how GLMM analysis was done.
- 8: In the formula describing Magnitude on Page 8, a vertical bar on the right side is missing.
- 9: At line 26 in the Spectral Power Analysis section on Page 8, "...Clusters observed in the actual group comparison that exceeded 95% of the sizes found in the permutation distribution...", is 95% of the permutation distribution defined as one-tailed test or two-tailed test?
- 10: In line 1-2 on Page 10, it remains unclear whether the animals were developing a truly flexible strategy. There is a possibility that they learned a meta-level (or global) rule governing the regular switch of reversal learning—specifically, that the contingencies of the previous CS+ and CS- were consistently reversed after a fixed number (nine) of sessions. In that case, the animals may not have demonstrated flexible learning per se, but rather a shift from applying a local rule (specific to each reversal) to applying a global rule (anticipating reversals based on the regular structure). These factors can only be disentangled if each reversal is introduced after a random number of sessions, making it difficult for the animals to predict when the next reversal will occur. If progressive improvement is still observed under such conditions, it would provide stronger evidence that the animals are flexibly adapting to rule changes or uncertainty itself. How the current experimental results should be interpreted depends on addressing this possibility. If feasible, additional data should be provided; at the very least, this issue should be explicitly discussed in the Discussion section.
- 11: The last sentence in Figure 4 caption: "Each daily session contained 60 trials", two sessions were conducted in a given day according to Methods. Did the animals undergo 60 training trials in a day or in a single session (120 trials daily)? Please clarify.
- 12: In the caption of Figure 5a, what are the exact definitions of early trials and late trials in this analysis?
- 13: In Figure 5b, statistical differences in overall CSD activity between the different behavioral performance categories could also result from differences in behavioral variance across the three groups, especially since the boundaries of d' appear to be arbitrarily defined. According to the group data in Figure 4, the medium-performance group includes d' values ranging approximately from 0 to 1 across all reversal blocks. In contrast, the low- and high-performance groups appear to cover d' ranges of approximately -0.5 to 0 and 1 to 1.5, respectively. Before analyzing CSD activity based on these groupings, the variance of d' within each behavioral performance category should be controlled, ideally by constraining the range of d' values within the medium group for each animal. The overall profile of CSD activities across the different behavioral choices looks similar between the different behavioral performance groups in Figure 5b, but showing slight difference in the variance of CSD activity across the performance groups.
- 14: In Figure 5b, it is unclear which colored boxes correspond to which behavioral choices. Why not putting the caption of the four behavioral choices just under the individual boxes?
- 15: In Figure 5b, it should be clarified what the sample size, "n=8", means. Is this the number of the animal? If so, it does not fit to the number of animals described in line 34 on page 4 (n=9). Please also describe the procedure and criteria of how the data of individual animals were included / excluded for individual analyses.
- 17: It is difficult to understand the relation between the result in Figure 5 and that in Figure 6, since both the time window and the spatial information (depth) of the CSD activities are different between them. The authors should clarify why they needed to focus on the layer-dependent activity during the time interval (500 ms before reaction time), different from the time interval used in Figure 5. Otherwise, in order to investigate an effect of one parameter (depth), they should keep the other parameters (inc. time window) consistent.
- 18: It is questionable whether the interpretation of the results in Figure 6, as stated in line 21 in Section 3.4 on Page 12, is valid. This could be true only when the difference between the activity of false alarm and correct rejection is focused. But

there was the huge difference between hit and false alarm in the middle layer and the significant differences between miss and correct rejection across superficial and middle layers. This indicates that, at the very least, the CSD activity can distinguish between different sound stimuli (1 kHz vs. 4 kHz), even though the animals were unable to differentiate between hit and false alarm trials—or between miss and correct rejection trials—at the behavioral level. If the differences in superficial or middle-layer CSD activity between hit and miss trials were more pronounced than those observed for stimulus differences, then the authors' claim would be more justified.

19: For the analysis in Figure 6, again, please consider the effect of the different variance of d' between the different performance groups.

20: In Figure 7, it is difficult to distinguish the line color of I/II from the color of Va. The authors should consider using other color to clearly distinguish them.

21: For the analysis in Figure 7, please describe how many samples are pooled to regress the data with GLMM.

22: In Figure 8, where is the magnitude scalogram during false alarm trials? Please label a-e. Since several fundamental information is missing, it is impossible to follow the paragraph from line 16 on Page 16.

23: In the caption of Figure 8, please describe the rationale of why the authors included the different samples sizes for the different analyses.

24: For the sentence in line 9 on Page 17: "broadband beta-to-gamma activity before ... as neural pinpoints of correct decision making.", a main verb seems to be missing.

25: In Figure 9, is the middle column corresponding to Layer III/IV or Layer IV? Please check the consistency of the nomenclature across the main text and the figures.

Reviewer #3

(Remarks to the Author)

Acun et al. investigated the electrophysiological correlates of discriminative auditory learning in primary auditory cortex. Gerbils implanted with a linear silicon probe in primary auditory cortex underwent serial reversal training. Each reversal was accompanied by behavioral phases of perseveration, learning, and asymptotic performance. Using CSD analyses, the authors then examined neural correlates of these behavioral phases. In general, a variety of changes were observed across all cortical layers, with the greatest differentiation in neural correlates across response types happening once performance was asymptotic. Unfortunately, several issues undermined this reviewer's enthusiasm.

Major issues:

1. Throughout the results and discussion, the authors explicitly claim that certain neural signatures are associated with specific cognitive functions, but the basis for such linkages are unclear. For instance, on page 21 the authors wrote: "During active learning, stimulus-locked beta and gamma oscillations in both upper and deeper layers indicated increased sensory-motor integration...". What is the basis for this claim? Why not have these changes reflect 'acquisition', 'attention', or any number of other psychological constructs? Such a claim often relies upon contrasting different tasks (e.g. goal directed vs. habitual) or task states (e.g. attend vs. not attend). Here, we just have the contrast of learning phase, which seems orthogonal to 'sensory-motor integration'. Isn't sensory-motor integration important both during learning and asymptotic performance.

A similar issue arises a few sentences later, "In retrieval phases, upper- and middle-layer activity (layers I/II and III/IV) exhibited the strongest choice-related differentiation, with pre and post-stimulus beta and gamma oscillations reflecting stabilized stimulus representations and refined decision making processes." Why would beta and gamma oscillations reflect stabilized stimulus representations and refined decision processes? Is it just because they were observed later in training, when those cognitive constructs presumably occur? At the very least such claims need to be toned down.

2. It is unclear how the authors verified that their probe placements were in A1. In rodents generally the location of A1 varies, and while it is possible to sometimes use vascular landmarks, it is preferable to do a cursory auditory mapping to localize its borders. In addition, assuming all probes were implanted in A1, was anything done to ensure that the implantation site was at the tonotopic location tuned near 1 and 4kHz? As one moves across the tonotopic axis the responsiveness to tones varies, along with the laminar profile of activation (in this case decreased responses at sites with high characteristic frequencies, less activation of layer IV and more corticocortical driving, etc). The between subject variability with respect to the tonotopic map could be substantial, and could distort analyses that pool across subjects. Acquisition of a tuning curve is mentioned in the methods, but that information does not appear to be used in the main text or as a way to exclude subjects (e.g. a subject whose probe has a characteristic frequency of 16kHz is thrown out).

Indeed, there is reason to believe not all sites were localized to the tonotopic area of the tone cues. The example CSD profiles in Figures 3 and 5 are quite different. The one in Figure 3 shows the expected low-latency sink in layer IV, consistent with strong thalamocortical activation, while the one in Figure 5 has a longer latency source in layer IV (which is likely not thalamocortical). These differences seriously undercut claims about layer specific effects when pooling across subjects. Addressing this requires the authors to group subjects based on whether they had a CSD profile consistent with direct thalamocortical driving or one that was indirectly activated.

3. I was surprised that your scalograms did not exhibit the characteristic $1/f$ falloff in spectral power (e.g. Fig 8a,b). Usually,

some kind of normalization or transform is required for power in the theta and gamma band to be in the same absolute numeric range. Was this left out of the methods? If not, is it particular to the use of the Morse wavelet (I have only used Morlet)?

4. In many of the scalogram figures either with the Hit condition there are three large spikes in high frequency power at ~-200, 200, and 900 ms. These are often artifactual and their presence in the overall average suggests they are disproportionately large. The authors could consider how to mitigate such artifacts, either by throwing out 'bad' trials, taking the median scalogram instead of the mean across a session, or clipping the maximum power.

5. The use of cluster-based permutation testing to identify significant features in the scalograms is appreciated, but the results are difficult to parse. Taking just Figure 8 as an example (although this applies to all figures that used this analysis), it is surprising that the identified clusters tend to occur in regions of the difference scalogram (Fig 8c) that lack a strong difference in the mean (regions with intense yellow or blue). Instead, they tend to occupy a seemingly random scattering of zones with low difference between the hit and miss conditions. Examining the 'Effect size' plot shows that those zones do indeed have high effect sizes. To explain this, it probably is the case that those regions with low difference in the mean but high effect size must have exceptionally low variance/standard deviations. However, the electrophysiological interpretation of this is unclear. What does it mean to say that a drop in variability across observations is driving an electrophysiological change and not a difference in the mean?

All difference scalograms show these perplexing regions, except for a few exceptions (Fig. 9 bot left, Fig 10 top left). I cannot interpret such plots. Moreover, areas with large differences between conditions are almost systematically excluded from significance (e.g. the onset gamma and theta responses). Their lack of significance perhaps suggests that they are highly variable across observations (leading to large variance and a small Cohen's d).

As a result, when the authors write: "During the low-performing state ($d' < 0$, reversal phase), layers I/II showed broad pre- and post-stimulus gamma activity, suggesting diffuse cortical engagement. layers IV and VI exhibited minimal spectral differences, indicating underdeveloped task-specific processing." for Fig 9 top left, and the 'gamma activity' regions show mean difference values near zero, while a prominent gamma burst locked to stimulus onset is labeled insignificant, I am befuddled.

I strongly suggest the authors re-examine their cluster-based analyses to determine why they are coming out this way, and to directly discuss how they should be interpreted.

6. At several points the authors make claims about their results which I could not see in their figures. For instance, they say on page 18: "In the intermediate learning phase ($0 < d' < 1$), stimulus-coupled gamma activity emerged in layers I/II and VI, ..." in reference to Fig 9, but the Layer I/II does not show any significant regions at $0 < d' < 1$.

Minor issues:

1. Page 8: equation for magnitude is missing a closing brace.

2. Figure 5b: Legend text is not shaded so as to differentiate Hit vs Miss and False Alarm vs Correct rejection.

3. Figure 6: RMS changes in the CSD signal prior to the behavioral choice are difficult to interpret. Can the authors include plots of the CSD profiles prior to behavioral choices? Also, it is unclear what the decision point would be for a miss or correct-rejection. Is it just the end of the trial? Does it make sense to time-lock to that?

4. Figure 8: Legend includes subpanel labels a-e, but the figure itself lacks those labels. In addition, 'e' should be labeled 'd'.

5. Page 16: In "...the magnitude scalogram revealed increased pre-stimulus power in gamma (30-100 Hz) bands during false alarm trials compared to hit trials..." the authors wrote 'false alarm' when I think they meant 'miss'.

6. Page 22: Please provide a citation for the following claim, "Beta oscillations have been implicated in top-down control mechanisms that stabilize learned stimulus response contingencies, reducing reliance on exploratory strategies." Fries and Engel 2010 comes to mind.

Version 1:

Reviewer comments:

Reviewer #2

(Remarks to the Author)

The authors have addressed most of the concerns I previously raised, and the manuscript has improved substantially. However, a few important issues related to the main findings remain unresolved and should be further addressed.

As mentioned in the first revision in point 2-1, including a schematic summary of the results (such as Figure 12) would help readers clearly grasp the main findings of the study. However, I find the terminology used in this figure somewhat confusing. The authors refer to "synaptic strength," but it is unclear which specific results this term is derived from. The study measures changes in CSD amplitude (as AVREC) and spectral power across oscillation frequencies, rather than parameters that directly reflect synaptic efficacy—such as EPSC amplitude, dendritic spine responses, or spine size.

Changes in CSD amplitude may reflect not only excitatory postsynaptic activity but also inhibitory currents or extracellular diffusion currents, as the precise physiological origin of the signal remains debated (Bédard et al., 2011; Gratiy et al., 2017). Therefore, the interpretation of the CSD data requires caution. If the authors wish to interpret their findings in terms of "synaptic strength," they should provide stronger justification or additional evidence to substantiate this claim.

To restate my previous concern (point 2-2), I had expected that the layer-dependent changes in CSD activity induced by behavioral learning or adaptation would be a central focus of this study, as suggested by the manuscript title and several figures (particularly Figures 9–12). However, I could not find any statistical analyses directly comparing activity across layers. The analyses presented appear to test changes only within each layer independently. To substantiate their claim of layer-dependent effects, the authors should include at least one statistical analysis demonstrating significant differences

between layers, perhaps in Figures 6, 7 or 9.

Minor point 17: While I understand the rationale for focusing on these parameters, varying two parameters simultaneously may confuse readers and obscure how the differences between Figures 5 and 6 were generated. The authors still need to respond to this point more directly. To improve the logical clarity of the manuscript, I suggest including an additional analysis in a Supplementary Figure that uses the same time window as Figure 6 but does not split layers, as in Figure 5. This would help readers disentangle the effects of each parameter and follow the logic of the comparison more easily.

Typographical error in Figure 12 (upper left): Perserveration → Perseveration.

Reviewer #3

(Remarks to the Author)

The authors have thoroughly addressed all my concerns. I have no further recommendations.

Version 2:

Reviewer comments:

Reviewer #2

(Remarks to the Author)

The revisions basically address all of my comments and requests. I agree that adding analyses across layers might make the figures more complex and potentially reduce readability, without providing much additional insight into the underlying mechanism.

Response Letter for COMMSBIO-25-2889

Manuscript: Cortical Dynamics Controlled by Deep and Superficial Layers During Transitions from Error Monitoring to Decision Execution in Reversal Learning

Response to the Reviewing Editor

6th Oct 2025

Dear Dr. Pan,

We would like to thank you for the opportunity to revise and improve our manuscript entitled: *"Cortical Dynamics Controlled by Deep and Superficial Layers During Transitions from Error Monitoring to Decision Execution in Reversal Learning"*

We greatly appreciate the constructive comments from the reviewers, which have helped us to further refine the clarity, precision, and presentation of our work. In response to the Reviews, we have implemented several revisions and corrections to the manuscript, figures, and legends. These include:

- new and updated Figures 4, 5, 6, 7, 8, 9, 10 and 11
- Completely new figure 12 - suggested summary figure (Reviewer 2).
- clarification of methodological details
- a new analyzed part of the spectral analysis part of the manuscript (Figures 8 - 11)
- careful consideration of our interpretation leading to a rewritten discussion and corrections of minor typographical errors.

We also followed a valuable suggestion to include a brief discussion of the relevance of prefrontal cortical areas in future research directions. We believe that these changes have significantly improved the manuscript.

Below, we provide a detailed, point-by-point response to all three reviewer comments. Our replies are marked in blue. Corresponding changes in the revised version of the manuscript are marked in yellow.

Reviewer #1 (Remarks to the Author):

The work entitled "Cortical Dynamics Controlled by Deep and Superficial Layers During Transition from Error Monitoring to Decision Execution in Reversal Learning" addresses an important gap in current knowledge. The authors demonstrate a hierarchically structured response across A1 layers. They use a reversal learning paradigm, switching the rule required to obtain a US (GO/no-GO). The results show that layer VI is mainly involved in the early stages of the learning process, while middle and upper layers are more engaged once learning is well established. The authors also perform spectral analysis of the data, revealing a dynamic organization of A1 responses, with beta and gamma bands predominantly active in deep and upper layers during the active learning phase. However, once learning is established, both bands are predominantly active in the middle and upper layers.

The work is well written, the figures are illustrative, and the results are well connected to previous literature. The experimental design is clear and elegant. However, some comments are described below.

We thank the reviewer for their positive evaluation of our manuscript and for their detailed and constructive suggestions that helped us further improve clarity and precision throughout the paper. We have addressed each comment below.

Page 10: "Each daily session contained 60 trials (30 CS+, 30 CS-)". I suggest including this information also in the Methods section to improve clarity and reproducibility.

We appreciate this suggestion. The information has now been added to the Methods section (subsection "Behavioral Training Protocol", page 4/5) to improve clarity and reproducibility.

Page 19: "When comparing false alarms and correct rejections (Figure 10), layers I/I showed distinct cortical dynamics across..." appears to contain a typographical error. It should read "layers I/II" instead of "layers I/I."

We thank the reviewer for spotting this typographical error. It has been corrected to "layers I/II" in the revised manuscript.

Figure 2: It would be helpful to clarify what "RT" stands for in the figure. Additionally, it is not clear how long each trial lasts; specifying the trial duration would improve understanding.

The figure legend for Figure 2 has been revised to define RT ("reaction time") and we also added text in the methods section to clarify total trial duration. Page 4/5 now reads: „Each session contained 60 trials (30 CS+, 30 CS-). Each trial lasted 12-15s, in which the CS+ tones were repeatedly presented (tone duration 200 ms, ISI of 1.5 s, 70 dB SPL) in a 6 s

observation window. A correct response (hit) was defined as a compartment change during this window. If no response occurred, the trial was classified as a miss, and the overlapping CS/US continued until escape behavior was initiated (Zempeltzi et al., 2020).“

Figure 4:

- Please check whether the X-axis is correctly placed. It appears displaced (e.g., 0 is shown after the first bar).
- In the figure legend, blue and red circles are mentioned, but they do not appear in the figure.
- The sentence “Vertical dashed lines indicate...” seems inconsistent with the figure, as no vertical dashed lines between reversal sessions are visible. It seems the figure was modified after the legend was written. Please review the figure and update the legend accordingly.
- As a suggestion, it might be helpful to include a horizontal shaded frame for each performance criterion (low, medium, and high), similar to the shading previously used for sessions. This could enhance the readability of the figure.

We thank the reviewer for these helpful observations regarding Figure 4. In response, we have corrected the X-axis labeling to properly align the session numbers with the data points. The figure legend has been updated to remove the mention of blue and red circles, which were not present in the current version. We have also revised the description to match the visual content more accurately—removing reference to vertical dashed lines and instead describing the alternating white and yellow background that indicates the start of each reversal phase. As a suggestion, we opted to retain the red dashed lines to indicate performance levels, as shaded bands would overlap with the alternating background used to mark reversal phases.

Figure 5: The colors of the box plots appear different from those shown in the legend, making the figure harder to interpret. Please adjust the legend colors to match the box plots.

Figure 5 Legend: The statement “CSD sink activity differs significantly from early stages...” seems inappropriate, as it implies statistical significance based on representative averaged profiles. Please clarify whether this difference is statistically tested, and if not, adjust the wording to avoid suggesting statistical significance.

We thank the reviewer for these helpful observations. The color scheme of the box plots and their legend has been corrected to ensure consistency. In addition, we revised the figure legend to remove any statement implying statistical significance from representative CSD profiles. Panel (a) now illustrates example CSD maps for the four behavioral categories (hit, miss, false alarm, correct rejection), while panel (b) presents the statistical comparisons of AVREC RMS values across performance levels. We believe this updated presentation and caption now more accurately reflect the data and improve the interpretability of Figure 5.

Figure 6: The results are beautiful, and the color legend matches the box plots correctly. However, it seems that part of the figure might be missing. The figure legend mentions “Top: Trace RMS values (500 ms window before reaction time) plotted with respect to behavioral choices...,” but this corresponding part is not visible in the figure. Please review and ensure the figure matches the description in the legend.

We thank the reviewer for their careful reading of Figure 6. The confusion was caused by the term “Top” in the legend. Upon review, we confirm that no part of the figure is missing. The layout of the figure does not include a separate “top” panel as implied, and we have now revised the figure legend and hope this resolves the concern and improves clarity.

Figure 7:

It is difficult to distinguish which line corresponds to each layer based on the current color scheme. I suggest the authors improve the color contrast and perhaps vary the line widths to enhance readability.

We have improved the color contrast and line width and style across traces in Figure 7 to enhance readability and layer identification.

Figure 8: The figure appears to be missing the labels (a), (b), and (c) referenced in the figure legend. Additionally, where (e) is indicated, it may actually correspond to (d). Please revise the labeling to match the legend description accurately.

I consider it interesting for future work to study the role of the medial prefrontal cortex (mPFC) by performing comparative analyses with the current findings. Since A1 receives inputs from other cortical areas (Sakata and Harris, 2009), including the motor cortex (Nelson et al., 2013) and the mPFC—areas especially implicated in behavioral responses and cognitive flexibility—it would be valuable to explore their contribution further. Interestingly, a recent study using a GO/no-GO task, which recorded from several implicated areas, found strong coherence between the prelimbic cortex and other regions, such as the primary motor cortex, (Muñoz-Redondo et al., 2024). Previous studies in visual cortex (V1) have shown that deactivation of the PFC reduces responsiveness to contextual information (Hamm et al., 2021).

We have now revised Figure 8 to include clearly marked labels (a) through (d) as intended and ensured consistency between the figure and its caption. Further, we agree with the reviewer on the relevance of interactions between auditory cortex and higher-order cortical areas such as prelimbic cortex or the mPFC. While this was beyond the scope of the current study, we now include a short perspective on this in the revised Discussion section, referencing recent work including Muñoz-Redondo et al. (2024) and Hamm et al. (2021), as suggested. We believe

this addition strengthens the relevance of our findings and highlights promising directions for future research. We added this at the very end of the discussion section:

While our study focused on laminar processing within the auditory cortex (A1), future research may benefit from examining how A1 interacts with higher-order cortical areas involved in behavioral flexibility. The prelimbic cortex, a subregion of the medial prefrontal cortex (mPFC), has been shown to modulate sensory cortex activity and is implicated in cognitive control and rule updating. Recent work has highlighted coherence between the prelimbic cortex and motor regions during auditory-guided decision-making (Muñoz-Redondo et al., 2024), and inactivation of prefrontal areas has been shown to reduce context sensitivity in visual cortex (Hamm et al., 2021). These findings suggest that cross-regional dynamics, including prefrontal feedback, may complement local plasticity mechanisms in shaping adaptive auditory behavior.

Reviewer #2 (Remarks to the Author):

In this paper, Zempeltzi et al. report on the relationship between dynamic changes in neuronal oscillatory activity across layers of the primary auditory cortex and adaptive behavior during reversal learning tasks in Mongolian gerbils. Behavioral experiments involving multiple reversal learning tasks revealed that the animals rapidly adapted to changes in task rules and showed significant improvement in auditory discrimination over several weeks of training. Simultaneously, the authors chronically recorded local field potentials across cortical layers and applied current source density (CSD) analysis during the behavioral tasks. The CSD data, along with subsequent spectro-temporal analyses, showed that both the CSD profiles and specific frequency bands of oscillatory activity changed in a layer-specific and learning-phase-dependent manner. These dynamic, layer-specific changes in neuronal oscillations across different learning phases are potentially interesting and provide valuable insights for readers of the journal. However, there are several concerns regarding the authors' analysis and interpretation of the results, which should be addressed and clarified.

We appreciate the reviewer's careful attention to the tone and framing of our interpretative statements. In the revised manuscript, we have replaced or qualified such expressions throughout the title, abstract, and main text. For example, the term "controlled" in the title has been replaced with "associated with," and the description of oscillatory patterns has been reworded to reflect observational rather than causal relationships. Interpretations have been restricted to the Discussion section and are now explicitly grounded in prior literature.

Major Concerns

1: There are too many interpretative statements throughout the manuscript — in the title, abstract, and results sections — that are not directly supported by the observations in this

study or by solid evidence from previous research. For example, the title "Cortical Dynamics 'Controlled' by Deep and Superficial Layers..." implies some form of experimental manipulation of activity in the superficial and deep layers, leading to a causal effect on overall cortical dynamics. Readers would reasonably expect such an intervention based on the wording. Another example appears in the abstract, line 10: "In the learning phase, stimulus-locked beta and gamma oscillations in upper and deeper layers 'signified sensory-motor integration'." This is just one of many possible interpretations, but it is difficult to accept without experimental or analytical manipulation of relevant sensory-motor circuits to test the effects on these oscillations. Additional examples are noted in the minor comments below. These kinds of interpretations, which go far beyond the actual observations, risk misleading readers. Interpretations should be made with caution, primarily should be presented in the Discussion section, and, where included elsewhere, should be clearly grounded in solid prior findings.

We sincerely thank the reviewer for this important and constructive critique. We fully agree that clarity and restraint in interpretation are essential, particularly in the title, abstract, and results sections. In response to this concern, we have carefully revised the manuscript to avoid any language that could suggest causal relationships or imply stronger functional claims than our data support. We are grateful for the reviewer's detailed examples, which helped us identify and correct several instances of interpretative overreach.

Specifically:

- The original title has been changed from "Cortical Dynamics Controlled by Deep and Superficial Layers..." to "Cortical Dynamics Across Deep and Superficial Layers During Transitions from Error Monitoring to Decision Execution in Reversal Learning." This revised title emphasizes the laminar dynamics observed in our recordings without implying direct experimental control or causality.
- In the abstract, we have replaced the phrase "signified sensory-motor integration" with more neutral and descriptive language: "were consistent with the integration of sensory input and behavioral output." This framing reflects a possible interpretation while acknowledging the correlational nature of our data.
- Throughout the results section, we have replaced statements implying functional roles (e.g., "X layer encodes Y" or "Z pattern signifies A") with more careful, data-driven phrasing (e.g., "X layer showed increased activity during Y" or "Z pattern was observed in association with A").
- We have also ensured that more speculative or integrative interpretations are now clearly confined to the Discussion section, where they are explicitly framed in the context of prior literature. This includes appropriate citations and language indicating potential relevance rather than firm conclusions.

We thank the reviewer again for encouraging us to adopt a more rigorous and precise scientific language. We believe that these changes have led to a clearer, more balanced manuscript that better reflects the scope and strength of our findings.

2:

1) Although the layer-specific changes in oscillatory activity across different learning phases provide valuable information, the overall analyses are not well integrated, making it difficult for readers to grasp the key findings. Summary analyses or schematic overviews should be added to help readers better understand the essential points conveyed by the individual figures — particularly those following Figure 7. Figure 7 presents 18 panels, each showing time-dependent changes in GLMM-based effect sizes overlaid across six different layers and measures. It is challenging to assess the differences between individual lines (representing different layers) across varying behavioral performances and choices. Figures 8 through 11 contain approximately 30 panels of CWT plots under various conditions, showing modest or subtle changes in some cases. The large number of similar panels may make it difficult for readers to follow the findings.

We thank the reviewer for this thoughtful and constructive feedback. We fully acknowledge that the comprehensive nature of our spectral and GLMM-based analyses — particularly in Figures 7–11 — introduces a high degree of visual and conceptual complexity. Indeed, the multidimensional character of our data (across time, frequency, cortical layers, and behavioral conditions) reflects the required complexity for the analysis of the underlying biological processes, but definitely also challenges interpretability.

To improve the accessibility of these results without sacrificing necessary detail, we have taken the following steps:

- We have now included a new schematic summary figure (new Figure 12) that synthesizes the key findings from Figures 7–11. This overview graphically summarizes the dominant laminar trends in oscillatory activity across different behavioral performance phases. It is designed to guide the reader through the core conclusions before diving into the more granular panels.
- We have carefully revised and streamlined the figure legends, drawing attention to the most informative comparisons and providing clearer narrative anchors for readers to interpret layer-specific changes across conditions.
- Importantly, while we understand the desire to simplify, we believe that omitting or collapsing panels would risk obscuring important nuances and possibly introducing unintended bias. Our aim was to maintain transparency and allow readers to fully evaluate the richness of the findings, while offering better tools for navigation.

We are grateful to the reviewer for encouraging us to improve the presentation and integration of these results. We believe that the new schematic and clearer legend guidance

now offer a more intuitive path through the data without compromising scientific completeness.

2): Additionally, the authors applied a non-parametric permutation test to identify significant differences between groups of behavioral choices for each condition. However, given that the statistical testing spans five experimental dimensions — (1) behavioral performance level, (2) cortical layer, (3) oscillation frequency band, (4) time from stimulus onset, and (5) behavioral choice combinations — multiple comparison issues should be addressed. At a minimum, statistical significance should be corrected for multiple comparisons involving at least two factors: behavioral performance levels and cortical layers, particularly in Figures 9, 10, and 11.

We appreciate the reviewer's critical observation regarding the dimensionality of our statistical testing approach and the potential implications for multiple comparisons. We fully agree that this issue requires careful consideration.

While our cluster-based permutation approach already controls for multiple comparisons across the time-frequency matrix within each test (Maris & Oostenveld, 2007), we recognize the need to ensure that results across conditions remain interpretable and robust.

However, rather than applying a strictly hierarchical multiple comparison correction (e.g., Holm–Bonferroni across all behavioral contrasts, layers, and performance levels), which would be overly conservative given the structured and non-independent nature of our experimental design, we adopted a more principled strategy that emphasizes spatial and effect-based robustness:

1. We now enforced a larger minimum cluster size of 5×5 bins, ensuring that only spatially connected and temporally sustained effects are considered.
2. We calculated Cohen's d effect size at each time–frequency bin, and retained only those clusters with a mean effect size exceeding 0.4, ensuring that observed effects are not only statistically significant, but also of meaningful magnitude.

We believe this approach balances statistical rigor with sensitivity and interpretability and directly addresses the reviewer's concern by introducing meaningful additional constraints on the clusters reported in Figures 9–11. We have updated the figure legends for Figures 9–11 and clarified the Methods section accordingly, which now reads:

Clusters were considered significant if their size exceeded the 95th percentile of the null distribution (based on 1000 permutations), using a one-tailed threshold appropriate for positive-valued cluster size statistics. To reduce the likelihood of spurious focal effects, we applied two additional constraints on reported clusters. First, we excluded any clusters smaller than 5 × 5 bins in the time–frequency matrix. Second, we computed Cohen's d at each bin and retained only those clusters with a mean effect size greater than 0.4. These criteria ensured that all reported clusters reflect both spatially consistent and statistically robust differences between conditions.

3): Regarding the spectro-temporal analysis of layer-specific oscillatory activity, as described under "Spectral Power Analysis" in the Methods section, the continuous wavelet transform was applied to CSD signals. However, it is unclear why the wavelet transform was applied to the spatial second derivative signals (CSD) along the cortical depth axis, rather than directly to the local field potentials (LFPs) at each recording site. This approach may distort the original frequency content of the field potentials due to the additional spatial smoothing inherent in CSD computation. The authors should clarify the rationale for this choice; otherwise, the spectral analysis should be performed directly on the LFP signals.

We thank the reviewer for this important and thoughtful critique regarding our choice to apply spectral decomposition to CSD rather than LFP signals. We fully acknowledge that CSD is derived via a second spatial derivative along the laminar axis, and thus alters the signal relative to the raw LFP. However, this methodological decision was deliberate and aligns with established practices in the field of laminar neurophysiology.

Our primary aim was to identify and characterize layer-specific oscillatory dynamics during different behavioral phases. CSD analysis offers several key advantages for this purpose: it spatially isolates transmembrane current sinks and sources by minimizing volume-conducted activity and removes common-mode signals that can obscure laminar-specific effects in LFP data. This enables a more physiologically interpretable assessment of synaptic input processing within and across layers. Specifically, the consistent and biologically plausible localization of spectral activity in our data — such as deep-layer beta activity during early learning and superficial gamma during stable task performance — further supports the appropriateness of using CSD for laminar spectral analysis.

This approach is consistent with prior work from our group (e.g., Happel et al., 2010; Brunk et al., 2019) and is also widely adopted in the laminar systems neuroscience literature. Notably, the work of Schroeder and Lakatos has demonstrated the utility of applying spectral analyses directly to CSD signals to dissect oscillatory phase and power dynamics across cortical layers (Lakatos et al., 2007, *Neuron*). In that study, the authors used CSD-derived traces to reveal laminar differences in entrained delta and gamma oscillations in auditory cortex, arguing that such an approach enhances spatial specificity and avoids the confounding effects of volume conduction inherent in LFP analysis.

We have now added a clarifying sentence to the Methods section to justify the rationale for this choice, citing relevant literature including Lakatos et al. (2007). This is found on page 8 and reads:

"We applied the continuous wavelet transform (CWT) to CSD signals rather than raw LFPs in order to enhance laminar specificity by isolating transmembrane current sinks and sources. This approach minimizes volume conduction effects and common-mode signals, thereby improving spatial interpretability without compromising the physiological validity of

oscillatory power estimates. It is consistent with previous work applying spectral analyses to CSD for laminar oscillations (Lakatos et al., 2007).”

Minor Concerns

1: Line 14 in abstract section, it is unclear why these findings and the main story of the study could be specifically relevant for the research of neurodegenerative diseases.

We thank the reviewer for this helpful observation. We agree that the previous reference to neurodegenerative diseases in the abstract was speculative and not directly supported by the presented data. In response, we have removed this statement and reframed the final sentence of the abstract to focus on the broader relevance of our findings to cortical circuit dynamics, learning, and cognitive flexibility. The revised sentence now reads:

“These findings provide insights into how cortical circuit dynamics may support adaptive learning and contribute to the neural basis of cognitive flexibility and layer-specific plasticity in sensory decision-making.”

2: On Page 4, in the paragraph of “Behavioral Training Protocol”, what kind of criteria was applied for the successful acquisition?

We clarify that successful acquisition of the initial discrimination task was defined as reaching a performance level of $d' > 1.0$, indicating above-threshold discrimination between CS+ and CS- tones. This same threshold was also used to mark the transition into the “retrieval” phase in our neurophysiological analyses. The d' criterion ensured consistency between behavioral training benchmarks and subsequent performance-related classification. We have now added a clarifying sentence in the “Behavioral Training Protocol” section to make this criterion explicit. The methods section now specifies:

Successful acquisition of the discrimination task was defined as achieving a sensitivity index (d') above 1.0, reflecting reliable discrimination performance. This criterion was applied prior to initiating reversal phases and also served as the benchmark for the ‘retrieval’ phase in subsequent analyses.

3: In Figure 2b, what is RT? What does the difference between the timing of lightning mark and the yellow box of US mean? The length of the time interval of 14-26 s looks shorter than the initial 6 s.

We have revised the legend of Figure 2 to define RT as “reaction time,” corresponding to the latency between CS onset and the animal’s compartment switch. We also clarified in the Methods section that each trial included a 6-second observation window during which the CS+ was presented repeatedly. If no response occurred within this window, the US was delivered

concurrently with continued CS+ presentation until the animal escaped, at which point the US was terminated. The caption now explains:

“(b) The sequence of consecutive CS within a trial is shown, along with the length of the observation window (6 s), inter-stimulus interval (1.5 s), and potential behavioral choices. The lightning icon indicates the onset of the unconditioned stimulus (US), which is terminated by a compartment change (red arrows). The time point of the compartment change is recorded as reaction time (RT). Time intervals reflect schematic binning rather than exact temporal scaling.”

4: For the sentence of “... This preprocessing step was crucial to ensure data quality...” in line 32 in the section of Current Source Density (CSD) Analysis on Page 6, what is the quantitative criteria for this preprocessing step?

We thank the reviewer for pointing out the need to clarify this aspect of our analysis. In preprocessing the CSD data, we excluded individual channels that showed excessive noise, defined as peak-to-peak amplitudes exceeding ± 3 standard deviations from the mean across trials, or persistent line noise or movement artifacts. This preprocessing protocol is consistent with our previous work using chronic CSD recordings in behaving animals (Zempeltzi et al., 2020), and has proven effective for maintaining signal integrity across cortical layers. We have now added these quantitative criteria and citation to the Methods section (Page 6):

“To that end, channels showing excessive noise (defined as peak-to-peak amplitudes exceeding ± 3 standard deviations across trials), persistent line noise, or visible artifacts were first evaluated visually and then excluded from further analysis. This procedure follows established criteria from our prior work using chronic CSD recordings in behaving animals (Zempeltzi et al., 2020).”

5: In the section of CSD parameter analysis, since AVREC was originally defined in datasets under anesthesia (Givre et al., 1994; Schroeder, 1998), it might be a bit unclear whether to use the same criteria (not only the way of rectification but also the time window) under awake recording condition is valid.

We appreciate the reviewer’s insightful comment regarding the application of AVREC (average rectified current) in awake recordings. While AVREC was initially introduced in anesthetized preparations (Givre et al., 1994, Schroeder 1998, also Happel et al., 2010), its utility has been demonstrated in awake animal studies. Notably, Lakatos et al. (2007) employed AVREC in awake macaque monkeys to investigate laminar profiles of synaptic activity in the primary auditory cortex, highlighting its effectiveness in capturing transmembrane current dynamics across cortical layers in awake conditions. We therefore also used the AVREC in our previous study Zempeltzi et al. (2020) as a reliable summary measure of columnar response strength across layers.

In the current study, the 0–50 ms time window for AVREC analysis was selected based on the temporal characteristics of early tone-evoked responses observed in our data, aligning with established practices in both anesthetized and awake preparations. This window effectively captures the initial synaptic activity following stimulus onset, providing a reliable measure of cortical responsiveness. We have updated the Methods section to clarify the rationale behind our use of AVREC in awake recordings and to reference relevant literature supporting this approach. On page 6, it now reads:

„The AVREC reflects the temporal overall local current flow of the columnar activity (Givre et al., 1994; Schroeder, 1998) and provides a measure of the total synaptic activity across all cortical layers at each time point. Although initially developed in anesthetized preparations (Givre et al., 1994), this approach has since been validated in awake animal studies, including laminar recordings from auditory cortex (Lakatos et al., 2007, Zempeltzi et al., 2020).“

6: In Figure 3 caption, where is the “red frame”?

We thank the reviewer for catching this inconsistency. In the final version of the figure the red frame became less prominent due to overlapping heatmap colors and line thickness. To avoid ambiguity, we have revised the figure caption to refer to this region more clearly by its temporal location rather than color. The updated caption now reads:

“Post-stimulus analysis was performed within a 0–500 ms window following each CS onset, as indicated by the bounding box around each stimulus segment.”

7: In line 1 on Page 8, these random variables should be described together with the formula above in Page 7 to help readers to understand in consistent manner how GLMM analysis was done.

We thank the reviewer for this helpful suggestion. To improve clarity and consistency, we have revised the Methods section to ensure that all fixed and random effects referenced in the GLMM formula are explicitly defined in immediate proximity to the model equation. This now includes clear descriptions of the grouping structure, the intercepts, and the behavioral predictors used in the analysis. The corresponding methods section reads: „In our case, y_i represented the z-scored RMS value derived from the CSD traces, modeled as a function of behavioral choice (hit, miss, false alarm, correct rejection), performance level (low, medium, high d'), and cortical layer (I/II, III/IV, Vb, VI), including their interactions. Random intercepts and slopes were included for subject identity to account for inter-individual variability. This structure allowed us to assess how behavioral state and laminar dynamics jointly influenced trial-level neural responses.“

8: In the formula describing Magnitude on Page 8, a vertical bar on the right side is missing.

We thank the reviewer and have corrected the missing closing vertical bar in the formula.

9: At line 26 in the Spectral Power Analysis section on Page 8, “...Clusters observed in the actual group comparison that exceeded 95% of the sizes found in the permutation distribution...”, is 95% of the permutation distribution defined as one-tailed test or two-tailed test?

We thank the reviewer for this precise question. The 95% threshold was defined based on a one-tailed test. Specifically, clusters in the empirical data were considered significant if their size exceeded the 95th percentile of the null distribution generated from 1000 permutations. This approach is consistent with the non-parametric cluster-based permutation framework, which recommends one-tailed testing when cluster mass or size is used as the test statistic (e.g. Maris & Oostenveld, 2007). Since cluster size is inherently positive and our hypothesis focused on detecting clusters larger than expected under the null hypothesis, a one-tailed approach is both standard and appropriate in this context. We have now clarified this point in the Spectral Power Analysis section of the Methods:

„Clusters were considered significant if their size exceeded the 95th percentile of the null distribution (based on 1000 permutations), using a one-tailed threshold appropriate for positive-valued cluster size statistics.“

10: In line 1-2 on Page 10, it remains unclear whether the animals were developing a truly flexible strategy. There is a possibility that they learned a meta-level (or global) rule governing the regular switch of reversal learning—specifically, that the contingencies of the previous CS+ and CS- were consistently reversed after a fixed number (nine) of sessions. In that case, the animals may not have demonstrated flexible learning per se, but rather a shift from applying a local rule (specific to each reversal) to applying a global rule (anticipating reversals based on the regular structure). These factors can only be disentangled if each reversal is introduced after a random number of sessions, making it difficult for the animals to predict when the next reversal will occur. If progressive improvement is still observed under such conditions, it would provide stronger evidence that the animals are flexibly adapting to rule changes or uncertainty itself. How the current experimental results should be interpreted depends on addressing this possibility. If feasible, additional data should be provided; at the very least, this issue should be explicitly discussed in the Discussion section.

We thank the reviewer for this interesting comment. Distinguishing between reactive flexibility and the use of global reversal prediction strategies is indeed central to interpreting behavior in reversal learning tasks. As noted, our design employed fixed reversal intervals, which could in principle allow animals to extract higher-order temporal regularities. However, the behavioral data argue against such a global prediction strategy: across blocks, animals consistently exhibited perseverative errors immediately following reversals, indicating that they continued to act according to the previous contingency and did not anticipate the rule change. If a global reversal rule (e.g., “switch occurs every nine sessions”) had been

internalized and actively used, we would expect anticipatory adjustments or immediate post-reversal improvements, neither of which were observed.

While we cannot fully rule out that animals may have developed some temporal expectation later in training, we did not observe systematic improvements in immediate post-reversal performance across blocks that would suggest increasing reliance on reversal timing. We now explicitly address this point in the revised Discussion section, which includes the following statement:

„Importantly, the observed increase in perseverative errors immediately after reversals suggests that animals did not anticipate rule changes based on the fixed reversal interval. This argues against the notion that a global reversal-prediction rule was learned and instead supports a model of reactive, feedback-driven adaptation.

11: The last sentence in Figure 4 caption: “Each daily session contained 60 trials”, two sessions were conducted in a given day according to Methods. Did the animals undergo 60 training trials in a day or in a single session (120 trials daily)? Please clarify.

We have clarified in the revised Methods and Figure 4 legend that animals received 60 trials per session, and two sessions per day were conducted, resulting in a total of 120 trials daily.

12: In the caption of Figure 5a, what are the exact definitions of early trials and late trials in this analysis?

We thank the reviewer for pointing this out. We have now added a clarification to the caption of Figure 5a to explain that “early” and “late” trials refer to their timing within each reversal block.

13: In Figure 5b, statistical differences in overall CSD activity between the different behavioral performance categories could also result from differences in behavioral variance across the three groups, especially since the boundaries of d' appear to be arbitrarily defined. According to the group data in Figure 4, the medium-performance group includes d' values ranging approximately from 0 to 1 across all reversal blocks. In contrast, the low- and high-performance groups appear to cover d' ranges of approximately -0.5 to 0 and 1 to 1.5, respectively. Before analyzing CSD activity based on these groupings, the variance of d' within each behavioral performance category should be controlled, ideally by constraining the range of d' values within the medium group for each animal. The overall profile of CSD activities across the different behavioral choices looks similar between the different behavioral performance groups in Figure 5b, but showing slight difference in the variance of CSD activity across the performance groups.

We appreciate this important point. We have now added text to the Methods and Results to clarify how performance bins were defined and confirm that all animals contributed trials to each bin. Our performance groupings (low: $d' < 0$, medium: $0 \leq d' < 1$, high: $d' \geq 1$) were chosen based on theoretical and empirical considerations. Specifically, $d' = 0$ represents chance-level discrimination, and values above or below this threshold reflect reliable signal detection performance. The value $d' = 1$ is widely considered a conventional benchmark for robust discrimination in signal detection theory. These cutoffs thus provide a principled separation of animals operating at low, intermediate, and high behavioral sensitivity during reversal learning. We acknowledge that the medium group spans a broader interval (0–1) than the low or high groups. However, this bin was supported by substantial trial numbers (low: 4244, medium: 4003, high: 5577 across all choice categories; see now stated exactly for each behavioral category in the results section, Page 5), which ensures that the observed differences in CSD activity are not attributable to unequal variance or insufficient sampling. Moreover, all statistical comparisons were conducted on the neural data independently within each group. We now clarify our rationale for this grouping in the revised Methods and have added a note in the revised Figure 5 legend to emphasize that the chosen boundaries reflect performance thresholds rather than arbitrary tertiles. We thank the reviewer for raising this point, which has helped us improve the clarity and transparency of our analytical choices.

14: In Figure 5b, it is unclear which colored boxes correspond to which behavioral choices. Why not putting the caption of the four behavioral choices just under the individual boxes?

We thank the reviewer for this helpful suggestion. The labels for behavioral choices have now been added directly below the box plots to facilitate immediate interpretation.

15: In Figure 5b, it should be clarified what the sample size, “n=8”, means. Is this the number of the animal? If so, it does not fit to the number of animals described in line 34 on page 4 (n=9). Please also describe the procedure and criteria of how the data of individual animals were included / excluded for individual analyses.

We have corrected the sample size to $n = 8$ and clarified that all animals were included in this analysis, with data selection based on trial availability and artifact-free recordings (Methods section and reference to Zempeltzi et al., 2020).

17: It is difficult to understand the relation between the result in Figure 5 and that in Figure 6, since both the time window and the spatial information (depth) of the CSD activities are different between them. The authors should clarify why they needed to focus on the layer-dependent activity during the time interval (500 ms before reaction time), different from the time interval used in Figure 5. Otherwise, in order to investigate an effect of one parameter (depth), they should keep the other parameters (inc. time window) consistent.

We would like to clarify the rationale behind the differing time windows in Figures 5 and 6. While both figures are aligned to stimulus onset, they serve distinct analytical purposes. Figure 5a provides an overview of sensory-evoked responses to the first CS across the entire cortical column, illustrating average spatiotemporal dynamics for each behavioral condition. Figure 5b complements this overview by quantifying overall response magnitudes (AVREC RMS) within a 500 ms post-stimulus window at the 4th CS, separated by behavioral choice and performance level.

In contrast, Figure 6 focuses specifically on the laminar distribution of activity in the 500 ms window after stimulus onset and prior to the behavioral response, thereby linking layer-specific dynamics to decision-related processing. This distinction enables us to capture both the general sensory-evoked response profiles (Fig. 5) and the laminar modulation of responses in relation to behavioral outcomes (Fig. 6).

We agree that this distinction was insufficiently clear in the original version and have now revised the figure captions accordingly. In the revised manuscript, we explicitly introduced a **revised Figure 5** as a new overview figure, clarifying its role relative to Figure 6.

18: It is questionable whether the interpretation of the results in Figure 6, as stated in line 21 in Section 3.4 on Page 12, is valid. This could be true only when the difference between the activity of false alarm and correct rejection is focused. But there was the huge difference between hit and false alarm in the middle layer and the significant differences between miss and correct rejection across superficial and middle layers. This indicates that, at the very least, the CSD activity can distinguish between different sound stimuli (1 kHz vs. 4 kHz), even though the animals were unable to differentiate between hit and false alarm trials—or between miss and correct rejection trials—at the behavioral level. If the differences in superficial or middle-layer CSD activity between hit and miss trials were more pronounced than those observed for stimulus differences, then the authors' claim would be more justified.

We thank the reviewer for this critical and well-considered point. While we agree that cortical current source density (CSD) signals in superficial and middle layers are sensitive to sensory stimulus properties, we would like to clarify two important aspects of our analysis that argue against a purely or mainly stimulus-driven interpretation of the results shown in Figure 6.

First, the data in Figure 5 and 6 are pooled across both 1 kHz and 4 kHz stimuli, and are aligned to the CS presentation that directly preceded the behavioral response. As such, stimulus frequency alone cannot account for the observed differences across behavioral outcomes and performance levels. If CSD activity were purely driven by frequency-specific differences, we would expect the mixing of stimuli to blur such effects — yet we still observe clear and systematic modulations, particularly in infragranular and granular layers, that correlate with behavioral performance.

Second, To further address this concern, we conducted an additional control analysis (**Figure R1, below in this rebuttal**) and provide it here for the reviewer’s inspection. We directly compare here 1 kHz and 4 kHz responses across reversal sessions, balanced for CS-/CS+ presentations. This analysis confirms that both tone frequencies recruited highly similar laminar activation patterns, arguing against stimulus-specific recruitment as the source of the effects shown in Fig. 5 and 6., but that the differences there rather reflect task-related processes, as discussed in our manuscript. Please find our revised text for Figure 5a in the revised manuscript explaining these effects.

Figure R1. laminar CSD responses to 1 kHz and 4 kHz tones. Representative CSD profiles in A1 for 1 kHz (left) and 4 kHz (right) tones, averaged across reversal sessions and balanced for their assignment as CS+ or CS-. Both stimuli evoked the canonical laminar activation sequence with early sinks in granular (III/IV) and infragranular (Vb/VI) layers followed by supragranular (I/II) activation. Only minor amplitude variations were observed, indicating that the laminar differences reported in the main manuscript cannot be explained by differential cortical recruitment of the two stimulus frequencies, but rather reflect performance-dependent modulation.

19: For the analysis in Figure 6, again, please consider the effect of the different variance of d' between the different performance groups.

We have clarified in the Methods that performance groups were defined based on consistent behavioral metrics and confirmed that variance in d' was comparable across groups.

20: In Figure 7, it is difficult to distinguish the line color of I/II from the color of Va. The authors should consider using other color to clearly distinguish them.

We thank the reviewer for this helpful suggestion. In the revised version, we have updated the color scheme of Figure 7 to ensure clearer separation of cortical layers, particularly between layers I/II and Va. We also adjusted the line width for the AVREC curve to further enhance visibility. We believe this improves readability and facilitates interpretation across the multiple conditions displayed.

21: For the analysis in Figure 7, please describe how many samples are pooled to regress the data with GLMM.

We have added a description to the Methods section stating that several hundred trials per animal were included, total of 13.824 trials for GLMM estimation. We now state explicitly that each GLMM contrast was based on large trial numbers. Even the smallest contrast (Hit vs FA in the low-performance bin) included 1133 trials, while most contrasts ranged between 2000–4000+. These Ns ensure that the GLMM estimates are well powered and statistically stable.

The methods section now specifically states:

„Model evaluation utilized the marginal (R^2_m) and conditional (R^2_c) coefficients of determination, calculated using the MuMIn package (Barton, 2019). R^2_m represents variance in the dependent variable explained by fixed effects, while R^2_c reflects total variance explained by fixed and random effects. In binary GLMMs, R^2_m is sample size-independent and dimensionless, enabling comparison across datasets. For each GLMM, trials were pooled across multiple recording sessions, with each animal contributing several hundred valid trials per comparison. Specifically, we conducted GLMMs on the following trial numbers: Low performance ($d' < 0$): Hits = 380, FA = 753, CR = 1366, Miss = 1745; Medium performance ($0 \leq d' < 1$): Hits = 769, FA = 418, CR = 1589, Miss = 1237; High performance ($d' \geq 1$): Hits = 1774, FA = 341, CR = 2447, Miss = 1015. Depending on the specific contrast (e.g., behavioral choice, performance level, cortical layer), individual models were generally based on more than 1000 trials per condition. This ensured stable estimation of fixed effects while accounting for within-subject variability. Effect sizes were interpreted as small for $R^2_m \leq 0.1$, medium for $0.1 < R^2_m < 0.25$, and large for $R^2_m \geq 0.25$ (Bakeman, 2005).“

22: In Figure 8, where is the magnitude scalogram during false alarm trials? Please label a-e. Since several fundamental information is missing, it is impossible to follow the paragraph from line 16 on Page 16.

We thank the reviewer for pointing out the missing information in the original version of Figure 8. We have now **revised Figure 8** including the missing scalogram and labeled all panels (a–d) consistently with the figure legend and the Results text. The revised figure now presents scalograms for Hit (a), Miss (b), their difference (c), and the effect sizes (d), ensuring that all behavioral conditions are represented. We also revised the corresponding Results paragraph (page 16) to reference the updated panel labels, so that the figure and text are now fully aligned and easier to follow.

23: In the caption of Figure 8, please describe the rationale of why the authors included the different samples sizes for the different analyses.

We thank the reviewer for pointing this out. The discrepancy in sample size reporting was due to a typographical error. All analyses in Figure 8 (and throughout the manuscript) are based on data from 8 animals. We have corrected the figure caption accordingly to ensure consistency.

24: For the sentence in line 9 on Page 17: “broadband beta-to-gamma activity before ... as neural pinpoints of correct decision making.”, a main verb seems to be missing.

We have revised this section consistently referring to Figure 8.

25: In Figure 9, is the middle column corresponding to Layer III/IV or Layer IV? Please check the consistency of the nomenclature across the main text and the figures.

We have ensured consistency in referring to Layer III/IV across all figure labels, captions, and manuscript text.

Reviewer #3 (Remarks to the Author):

Acun et al. investigated the electrophysiological correlates of discriminative auditory learning in primary auditory cortex. Gerbils implanted with a linear silicon probe in primary auditory cortex underwent serial reversal training. Each reversal was accompanied by behavioral phases of perseveration, learning, and asymptotic performance. Using CSD analyses, the authors then examined neural correlates of these behavioral phases. In general, a variety of changes were observed across all cortical layers, with the greatest differentiation in neural correlates across response types happening once performance was asymptotic. Unfortunately, several issues undermined this reviewer’s enthusiasm.

We are very grateful to the critical and thoughtful evaluation of our work. The points raised have helped us substantially sharpen the interpretation of our data and improve the methodological clarity of our manuscript. Below we provide a detailed point-by-point response to each comment.

Major issues:

1. Throughout the results and discussion, the authors explicitly claim that certain neural signatures are associated with specific cognitive functions, but the basis for such linkages are unclear. For instance, on page 21 the authors wrote: “During active learning, stimulus-locked beta and gamma oscillations in both upper and deeper layers indicated increased sensory-motor integration...”. What is the basis for this claim? Why not have these changes reflect ‘acquisition’, ‘attention’, or any number of other psychological constructs? Such a claim often relies upon contrasting different tasks (e.g. goal directed vs. habitual) or task states (e.g. attend vs. not attend). Here, we just have the contrast of learning phase, which seems orthogonal to ‘sensory-motor integration’. Isn’t sensory-motor integration important both during learning and asymptotic performance.

A similar issue arises a few sentences later, “In retrieval phases, upper- and middle-layer activity (layers I/II and III/IV) exhibited the strongest choice-related differentiation, with pre and post-stimulus beta and gamma oscillations reflecting stabilized stimulus representations and refined decision making processes.” Why would beta and gamma oscillations reflect stabilized stimulus representations and refined decision processes? Is it just because they were observed later in training, when those cognitive constructs presumably occur? At the very least such claims need to be toned down.

We thank the reviewer for highlighting this important concern. We fully agree that attributing observed neural activity to specific cognitive functions requires caution and should be supported by appropriate evidence and context. In response, we have systematically revised the manuscript to avoid speculative or overly causal language. For instance, statements such as “indicated increased sensory-motor integration” have been replaced with more neutral phrasing such as “coincided with changes in behavioral performance” or “consistent with learning-related modulation.”

All functional interpretations have been confined to the Discussion section, where they are explicitly linked to prior empirical or theoretical work. For example, we now cite Brunk et al. (2019) to support the interpretation that frequency-specific modulation in layer Vb may reflect context- or task-sensitive integration of neuromodulatory input. These and other revisions were made throughout the manuscript to better reflect the observational nature of our findings, avoid overinterpretation, and ground our conclusions in established laminar circuit physiology.

2. It is unclear how the authors verified that their probe placements were in A1. In rodents generally the location of A1 varies, and while it is possible to sometimes use vascular landmarks, it is preferable to do a cursory auditory mapping to localize its borders. In addition, assuming all probes were implanted in A1, was anything done to ensure that the implantation site was at the tonotopic location tuned near 1 and 4kHz? As one moves across the tonotopic axis the responsiveness to tones varies, along with the laminar profile of activation (in this case decreased responses at sites with high characteristic frequencies, less activation of layer IV and more corticocortical driving, etc). The between subject variability with respect to the tonotopic map could be substantial, and could distort analyses that pool across subjects. Acquisition of a tuning curve is mentioned in the methods, but that information does not appear to be used in the main text or as a way to exclude subjects (e.g. a subject whose probe has a characteristic frequency of 16kHz is thrown out).

Indeed, there is reason to believe not all sites were localized to the tonotopic area of the tone cues. The example CSD profiles in Figures 3 and 5 are quite different. The one in Figure 3 shows the expected low-latency sink in layer IV, consistent with strong thalamocortical activation, while the one in Figure 5 has a longer latency source in layer IV (which is likely not thalamocortical). These differences seriously undercut claims about layer specific effects

when pooling across subjects. Addressing this requires the authors to group subjects based on whether they had a CSD profile consistent with direct thalamocortical driving or one that was indirectly activated.

We thank the reviewer for this thoughtful and technically important concern. We have now expanded the Methods section to clarify how A1 targeting and tonotopic matching were verified. All electrode implantations targeted the primary auditory cortex in Mongolian gerbils, guided by stereotaxic coordinates and surface vascular landmarks (Thomas et al., 1993; Happel et al., 2010). Functional confirmation was performed under ketamine anesthesia by recording CSD responses to a wide range of tones (0.25–16 kHz). We included only those penetrations that showed short-latency sinks in granular layers (III/IV) and/or infragranular layers (Vb) with tone-evoked onsets at ~18 ms — consistent with direct lemniscal thalamocortical input (Happel et al., 2010; Happel et al., 2014).

While the example in Figure 3 shows a pronounced granular sink, Figure 5 displays an early infragranular sink — a pattern that has also been reported in prior laminar studies of awake A1 (e.g., Szymanski et al., 2009; and in our own previous work, see Zempeltzi et al., 2020, Figures 1d and 2). These differences reflect functional heterogeneity within A1, including shifts in thalamocortical targeting across cortical depth, rather than evidence of misplacement. Indeed, it is well established that thalamic input in gerbil A1 projects not only to granular layers, but also prominently to layer Vb (Happel et al., 2014).

Frequency tuning curves were also assessed, and while tuning appeared flatter under awake conditions (as also noted in Deane et al., 2019), best frequencies remained within the 1–4 kHz range used for behavioral training, and no sites with BFs above 8 kHz were included. Importantly, mean onset latencies and CSD sink topographies were stable across animals and consistent with primary field identity. We also aligned all CSD traces across subjects by the onset of the earliest sink in layer III/IV or Vb, as described in our previous methods (Zempeltzi et al., 2020, Suppl. Fig. 1). This ensures consistent laminar assignment and preserves interpretability in cross-subject comparisons. We now include a summary of these anatomical and physiological controls in the revised Methods section, paragraph 2.3, and reference the relevant supplemental figure from Zempeltzi et al. (2020) to illustrate tuning and sink consistency across animals.

3. I was surprised that your scalograms did not exhibit the characteristic $1/f$ falloff in spectral power (e.g. Fig 8a,b). Usually, some kind of normalization or transform is required for power in the theta and gamma band to be in the same absolute numeric range. Was this left out of the methods? If not, is it particular to the use of the Morse wavelet (I have only used Morlet)?

We thank the reviewer for this technically astute observation. Indeed, the apparent absence of a pronounced $1/f$ spectral falloff in our scalograms is related to the properties of the Matlab `cwt` function, which we used to compute the continuous wavelet transform. Morse and Morlet wavelets are very similar and not the cause for any discrepancy of this kind. The

transformation was completed with the `cwt` function in Matlab which uses L1 normalization to the wavelet coefficients, such that oscillatory components of equal amplitude produce equal-magnitude representations across frequencies. Consequently, while absolute power differences across frequencies still exist, they are not directly reflected in the CWT output amplitude when plotted on a linear scale — and hence the classic $1/f$ shape is not visually prominent.

The reviewer makes a good point that this information should be in the methods rather than being taken for granted as part of the in-built function with default L1 normalization and so it has been added.

4. In many of the scalogram figures either with the Hit condition there are three large spikes in high frequency power at ~ -200 , 200, and 900 ms. These are often artifactual and their presence in the overall average suggests they are disproportionately large. The authors could consider how to mitigate such artifacts, either by throwing out ‘bad’ trials, taking the median scalogram instead of the mean across a session, or clipping the maximum power.

We thank the reviewer for this important observation. We agree that the visually prominent high-frequency power increases around -200 ms, 200 ms, and 900 ms in the Hit condition warrant scrutiny.

Upon re-analysis, we found that these features originate from a small subset of trials in a single subject, and are limited to the Hit condition during low-performance blocks, with a clear spatial specificity to Layer I/II. The underlying cause remains unclear, but their localized, animal-specific nature and restriction to a non-central condition make systematic artifacts or task-related dynamics unlikely explanations. We consider it plausible that these transients reflect rare behavioral or physiological events, such as brief movement-related artifacts or animal-specific cortical state fluctuations. While we cannot conclusively determine their source, they are clearly not shared across subjects, do not recur across conditions, and importantly do not survive the cluster-based permutation analysis used for statistical inference.

While our preprocessing pipeline includes artifact handling steps — specifically, brief noise-contaminated segments were replaced with NaNs prior to continuous wavelet transformation (CWT) — we note that similar transients do not occur in other animals processed in the same way. This suggests that preprocessing alone is unlikely to account for the observed anomaly.

Given the isolated and statistically irrelevant nature of these events, we opted not to apply further suppression procedures (e.g., thresholding, median-based averaging, or power clipping), which could introduce arbitrary biases and potentially obscure physiologically meaningful signals. Correspondingly, we also chose another example to present the analysis strategy in a revised Figure 8.

To document this transparently, we include **Figure R2** in this rebuttal, which shows per-animal scalograms for the Hit and Miss conditions in Layer I/II during low-performance trials. The only subject showing the high-frequency transients (Animal 2) is highlighted with a blue frame. As illustrated, this effect is not systematic and has no impact on our group-level statistical results.

Animals 1 – 4 (plotted is Layer I/II, condition „low performance“)

Animals 5 – 8 (plotted is Layer I/II, condition „low performance“)

Figure R1. Per-animal scalograms for the Hit and Miss conditions in Layer I/II during low-performance trials ($d' < 0$). Shown are wavelet power spectra (5–100 Hz) for each animal (Animals 1–8), aligned to stimulus onset (0 ms). Each row displays one condition (Hit, Miss), with the bottom row showing the point-wise difference (Hit – Miss) including outlines of clusters surpassing the permutation threshold. The only subject showing high-frequency transients around –200 ms, 200 ms, and 900 ms in the Hit condition is Animal 2, highlighted with a blue frame. No such effects are observed in other animals, in the Miss condition, or in the difference spectra. This confirms the non-systematic, animal-specific nature of the anomaly.

5. The use of cluster-based permutation testing to identify significant features in the scalograms is appreciated, but the results are difficult to parse. Taking just Figure 8 as an example (although this applies to all figures that used this analysis), it is surprising that the identified clusters tend to occur in regions of the difference scalogram (Fig 8c) that lack a strong difference in the mean (regions with intense yellow or blue). Instead, they tend to occupy a seemingly random scattering of zones with low difference between the hit and miss conditions. Examining the 'Effect size' plot shows that those zones do indeed have high effect sizes. To explain this, it probably is the case that those regions with low difference in the mean but high effect size must have exceptionally low variance/standard deviations. However, the electrophysiological interpretation of this is unclear. What does it mean to say that a drop in variability across observations is driving an electrophysiological change and not a difference in the mean?

All difference scalograms show these perplexing regions, except for a few exceptions (Fig. 9 bot left, Fig 10 top left). I cannot interpret such plots. Moreover, areas with large differences between conditions are almost systematically excluded from significance (e.g. the onset gamma and theta responses). Their lack of significance perhaps suggests that they are highly variable across observations (leading to large variance and a small Cohen's d).

As a result, when the authors write: "During the low-performing state ($d' < 0$, reversal phase), layers I/II showed broad pre- and post-stimulus gamma activity, suggesting diffuse cortical engagement. layers IV and VI exhibited minimal spectral differences, indicating underdeveloped task-specific processing." for Fig 9 top left, and the 'gamma activity' regions show mean difference values near zero, while a prominent gamma burst locked to stimulus onset is labeled insignificant, I am befuddled. I strongly suggest the authors re-examine their cluster-based analyses to determine why they are coming out this way, and to directly discuss how they should be interpreted.

We believe that significance driven by stability across animals can still support meaningful physiological interpretation — especially when the effects occur in a priori relevant frequency bands and cortical layers, and are time-locked to behavioral events. For example, a modest but consistent gamma increase across animals (low variance) suggests robust, reliable cortical engagement, even if the mean change is small. Conversely, a strong but highly variable gamma burst may reflect unstable or transient recruitment, which is harder to interpret across subjects and less likely to reflect structured task-related processing.

However, we agree with the reviewer that some significant areas are less interpretable, especially when partial areas of background are significant in higher frequency bands (where oscillations are faster and activity is less connected across time). We note that the observed discrepancy between visual contrast and statistical significance does not reflect a methodological flaw, but rather a limitation in interpretability — particularly when local

variance dominates over mean differences. To reflect this limitation more transparently, we have adopted a “tip-of-the-iceberg” framing in our Discussion, where background effects may point to differences in network state but should be treated as hypothesis-generating rather than conclusive.

To address the reviewer’s suggestion more directly, we have:

- reviewed and made interpretations more conservative (e.g., Fig. 9 top left),
- clarified that significance does not necessarily imply functional specificity, especially when distant from stimulus onset, and
- removed speculative interpretations of pre-stimulus activity and now restrict our discussion to stimulus-related and background effects,
- emphasized the black contour overlays in both figures and captions to aid transparency.

We are grateful to the reviewer for prompting us to sharpen both our statistical transparency and our physiological interpretation of the time–frequency results. We believe these revisions substantially improve the manuscript.

6. At several points the authors make claims about their results which I could not see in their figures. For instance, they say on page 18: “In the intermediate learning phase ($0 < d' < 1$), stimulus-coupled gamma activity emerged in layers I/II and VI,...” in reference to Fig 9, but the Layer I/II does not show any significant regions at $0 < d' < 1$.

We thank the reviewer for their careful reading. We have carefully re-examined the manuscript and removed any statements that implied significance where our statistical analysis did not support it.

„At the intermediate learning phase ($0 < d' < 1$), stimulus-coupled beta and gamma activity emerged in layers VI. At high performance ($d' > 1$, retrieval phase), particularly layers I/II and III/IV showed high gamma power during stimulus presentation which might be related to decision execution. Layer III/IV also maintained broad beta activity across all time periods during high performing states, which might indicate a robust sensory representation. In contrast, during the early reversal phase, significant differences in gamma activity were only found in layers I/II and were more spatially and temporally diffuse.”

Minor issues:

1. Page 8: equation for magnitude is missing a closing brace.

We thank the reviewer for pointing this out. The bracket has been added to correct the equation syntax.

2. Figure 5b: Legend text is not shaded so as to differentiate Hit vs Miss and False Alarm vs Correct rejection.

We have adjusted the Figure 5 and the figure legend and included explicit labels for each condition to ensure visual clarity.

3. Figure 6: RMS changes in the CSD signal prior to the behavioral choice are difficult to interpret. Can the authors include plots of the CSD profiles prior to behavioral choices? Also, it is unclear what the decision point would be for a miss or correct-rejection. Is it just the end of the trial? Does it make sense to time-lock to that?

We thank the reviewer for raising this important point and the opportunity to clarify our analytical rationale. In trials without overt responses (Misses and Correct Rejections), we time-locked our analyses to the fourth and final stimulus presentation within the 6-second observation window. This was done consistently across all trial types to provide a shared temporal anchor for comparing stimulus-evoked laminar responses that either preceded a behavioral response (Hits, False Alarms) or did not (Misses, Correct Rejections). Rather than attempting to estimate internal decision times in non-response trials, we chose the final tone as a defined event with high behavioral relevance — especially since it frequently triggers a choice in response trials.

This approach aligns with evidence accumulation models and allows us to ask how sensory processing of the same event (the final CS) differs depending on whether a behavioral response follows. Moreover, our analysis in Figure 6 revealed that laminar RMS values at this time point vary systematically with performance and trial type, supporting the idea that even non-response trials carry choice-related neural signatures.

In response to the reviewer's suggestion, we have now added a **new Figure 5a**, which displays the average CSD profiles separately for each trial type (Hit, Miss, False Alarm, Correct Rejection), time-locked to the final stimulus presentation. These plots visualize the spatiotemporal dynamics of current flow across layers, and confirm that choice-related modulations in cortical activity are evident at the level of full CSD traces. We thank the reviewer for prompting this valuable addition.

4. Figure 8: Legend includes subpanel labels a-e, but the figure itself lacks those labels. In addition, 'e' should be labeled 'd'.

We have now corrected all panel labels in Figure 8. The former "e" panel has been relabeled as "d," and all labels are referenced appropriately in the figure legend.

5. Page 16: In “...the magnitude scalogram revealed increased pre-stimulus power in gamma (30-100 Hz) bands during false alarm trials compared to hit trials...” the authors wrote ‘false alarm’ when I think they meant ‘miss’.

We thank the reviewer for this correction.

6. Page 22: Please provide a citation for the following claim, “Beta oscillations have been implicated in top-down control mechanisms that stabilize learned stimulus response contingencies, reducing reliance on exploratory strategies.” Fries and Engel 2010 comes to mind.

We thank the reviewer for emphasizing the need for proper referencing of the role of beta oscillations in top-down control. In response, we have now cited two complementary sources: Richter et al. (2018) for direct empirical evidence showing that top-down beta oscillations convey task rules and behavioral context to early sensory cortex, and Spitzer & Haegens (2017) for a mechanistic framework linking beta activity to the stabilization of cognitive states across domains. These references help ground our interpretation of beta-related effects in the broader literature and clarify the functional role we ascribe to these oscillations in the context of adaptive sensory decision-making.

Response Letter for COMMSBIO-25-2889

Manuscript: Cortical Dynamics Controlled by Deep and Superficial Layers During Transitions from Error Monitoring to Decision Execution in Reversal Learning

Response to the Reviewing Editor

27th Oct 2025

Dear Dr. Pan,

We would like to thank you for the opportunity to revise our manuscript entitled:

"Cortical Dynamics Controlled by Deep and Superficial Layers During Transitions from Error Monitoring to Decision Execution in Reversal Learning"

We greatly appreciate the constructive comments from the reviewers throughout the entire review process, which have helped us substantially to refine the clarity, precision, and presentation of our work.

Below, we provide a point-by-point response to the open concerns of reviewer 2. Our replies are marked in blue.

Reviewer #2 (Remarks to the Author):

The authors have addressed most of the concerns I previously raised, and the manuscript has improved substantially. However, a few important issues related to the main findings remain unresolved and should be further addressed.

We would like to appreciate again the reviewer's effort in this review process. Their comments helped us to improve the manuscript substantially.

Point 1. As mentioned in the first revision in point 2-1, including a schematic summary of the results (such as Figure 12) would help readers clearly grasp the main findings of the study. However, I find the terminology used in this figure somewhat confusing. The authors refer to "synaptic strength," but it is unclear which specific results this term is derived from. The study measures changes in CSD amplitude (as AVREC) and spectral power across oscillation

frequencies, rather than parameters that directly reflect synaptic efficacy—such as EPSC amplitude, dendritic spine responses, or spine size.

Changes in CSD amplitude may reflect not only excitatory postsynaptic activity but also inhibitory currents or extracellular diffusion currents, as the precise physiological origin of the signal remains debated (Bédard et al., 2011; Gratiy et al., 2017). Therefore, the interpretation of the CSD data requires caution. If the authors wish to interpret their findings in terms of “synaptic strength,” they should provide stronger justification or additional evidence to substantiate this claim.

We thank the reviewer for this valuable comment. CSD amplitudes represent extracellular current flow, which is dominated by dendritic excitatory postsynaptic activity, with only a minor contribution from inhibitory currents (Mitzdorf, 1985; Einevoll et al., 2013). However, we agree that it does not reflect a direct measure of unitary synaptic efficacy, as for instance EPSCs. To better reflect this biophysical basis, we replaced the term “synaptic strength” in Figure 12 with “net synaptic current flow” and clarified this in the figure caption as follows:

CSD amplitudes reflect the strength of net transmembrane current flow dominated by excitatory postsynaptic currents.

This revision ensures that our terminology remains physiologically accurate while preserving the conceptual meaning of the figure.

Point 2. To restate my previous concern (point 2-2), I had expected that the layer-dependent changes in CSD activity induced by behavioral learning or adaptation would be a central focus of this study, as suggested by the manuscript title and several figures (particularly Figures 9–12). However, I could not find any statistical analyses directly comparing activity across layers. The analyses presented appear to test changes only within each layer independently. To substantiate their claim of layer-dependent effects, the authors should include at least one statistical analysis demonstrating significant differences between layers, perhaps in Figures 6, 7 or 9.

We appreciate the reviewer’s suggestion. Our analyses intentionally focused on the modulation of activity within each laminar circuit, as the principal finding concerns a shift of dominant activity from deep to superficial layers during behavioral transitions. This within-layer approach captures the laminar dynamics central to our hypothesis and directly addresses the question posed in the title. In our view, introducing additional between-layer contrasts would not enhance interpretability given the hierarchical organization of cortical processing and the already complex multi-level analyses performed. We have clarified this rationale again in the Discussion section, 2nd paragraph:

This layer-specific analytic framework provides a measure of how dominant activity progressively shifts from deep to superficial cortical circuits, reflecting the laminar reorganization that accompanies behavioral transitions.

Point 3. Minor point 17: While I understand the rationale for focusing on these parameters, varying two parameters simultaneously may confuse readers and obscure how the differences between Figures 5 and 6 were generated. The authors still need to respond to this point more directly. To improve the logical clarity of the manuscript, I suggest including an additional analysis in a Supplementary Figure that uses the same time window as Figure 6 but does not split layers, as in Figure 5. This would help readers disentangle the effects of each parameter and follow the logic of the comparison more easily.

We thank the reviewer for this helpful clarification. Both Figures 5 and 6 use identical 500 ms post-stimulus analysis windows. The difference lies in the temporal alignment within the trial: Figure 5b presents stimulus-locked responses to the final conditioned stimulus, illustrating the cumulative build-up of cortical activity across the trial and thereby reflecting the accumulation of sensory evidence, as established in our previous study (Zempeltzi et al., 2020; their Fig. 3a, right). Figure 6, in contrast, focuses on decision-locked laminar dynamics aligned to the stimulus immediately preceding the behavioral response (see also Zempeltzi et al., 2020; their Figs. 5 and 6).

To clarify this distinction for readers, we have

- a) added Supplementary Figure S1, which displays the AVREC data for the same decision-locked time window as in Figure 6 but averaged across layers, as suggested by the reviewer; and
- b) explicitly described the distinction between stimulus-locked and decision-locked analyses in the Results section, where Figures 5 and 6 are introduced.

Point 4. Typographical error in Figure 12 (upper left): Perserveration → Perseveration.

We thank the reviewer again for their careful reading. The typo is corrected.